# An Improved Generalized Chirp Scaling Algorithm Based on Lagrange Inversion Theorem for High-Resolution Low Frequency Synthetic Aperture Radar Imaging

**Xing Chen, Tianzhu Yi, Feng He, Zhihua He and Zhen Dong \***

College of Electronic Science, National University of Defense Technology, No. 109 De Ya Road, Changsha 410073, China

\* Correspondence: dongzhen@nudt.edu.cn; Tel.: +86-156-1622-6428

**Abstract:** The high-resolution low frequency synthetic aperture radar (SAR) has serious range-azimuth phase coupling due to the large bandwidth and long integration time. High-resolution SAR processing methods are necessary for focusing the raw data of such radar. The generalized chirp scaling algorithm (GCSA) is generally accepted as an attractive solution to focus SAR systems with low frequency, large bandwidth and wide beam bandwidth. However, as the bandwidth and/or beamwidth increase, the serious phase coupling limits the performance of the current GCSA and degrades the imaging quality. The degradation is mainly caused by two reasons: the residual high-order coupling phase and the non-negligible error introduced by the linear approximation of stationary phase point using the principle of stationary phase (POSP). According to the characteristics of a high-resolution low frequency SAR signal, this paper firstly presents a principle to determine the required order of range frequency. After compensating for the range-independent coupling phase above 3rd order, an improved GCSA based on Lagrange inversion theorem is analytically derived. The Lagrange inversion enables the high-order range-dependent coupling phase to be accurately compensated. Imaging results of P- and L-band SAR data demonstrate the excellent performance of the proposed algorithm compared to the existing GCSA. The image quality and focusing depth in range dimension are greatly improved. The improved method provides the possibility to efficiently process high-resolution low frequency SAR data with wide swath.

**Keywords:** synthetic aperture radar (SAR); low frequency; high-resolution; large bandwidth; improved generalized chirp scaling (GCS); Lagrange inversion theorem; range-dependent coupling

## 1. Introduction

Higher spatial resolution is an important development direction of synthetic aperture radar (SAR). Recent SAR systems are capable of resolutions in the decimeter regime. This requires the usage of large range bandwidth and wide azimuth beamwidth. The high-resolution, together with the all-weather day-and-night imaging capabilities, is turning SAR into an ideal tool for regular mapping and monitoring applications [1,2]. Moreover, microwaves can penetrate into vegetation and even the ground up to a certain depth [3]. The penetration capabilities depend on the carrier frequencies as well as on the complex dielectric constants, densities and conductivities of the observed targets. The high frequencies, like the X-band (8–12 GHz), show typically a high attenuation and are mainly backscattered on the top of the vegetation. Low frequencies, like P- and L-band (0.23~1 GHz and 1~2 GHz, respectively) [4], usually penetrate deep into vegetation, snow and ice. A high-resolution low frequency SAR system refers to a SAR system which operates with a low frequency (P- or

L-band) signal with a large fractional bandwidth (>0.2, i.e., the ultra-wideband SAR [2,5–8]) and a wide antenna beamwidth (corresponding to high azimuth resolution in the decimeter regime). The fractional bandwidth is defined by the ratio of the signal bandwidth to the center frequency. The combination of low frequency with large bandwidth and wide beam allows SAR to obtain high-resolution images of concealed targets, with the capability of penetrating the ground or foliage surface, thus it has broad applications for both military and civil purposes in recent years [9–12]. However, the large fractional bandwidth and long azimuth integration time used in high-resolution low frequency SAR bring new challenges to get high-quality images by the conventional image formation.

A crucial problem is the serious coupling between the range and azimuth frequencies in the phase of high-resolution low frequency SAR transfer function [13]. In two-dimension (2D) frequency domain, the phase is range-dependent and can be decomposed into two parts: the range-independent terms and the range-dependent terms. Different algorithms make different approximations of these two parts. At low frequencies, many of the simplifying assumptions made by traditional algorithms are not valid, such as range-Doppler algorithm [14] and chirp scaling algorithm (CSA) [15], resulting in serious image degradations as blurring and resolution loss. The problem stems from high-order range-azimuth phase coupling. Several approaches have been used in the processing of this type of SAR system. The time-domain algorithms and the wavenumber-domain Omega-K $(\omega - k)$ algorithm [16–18] are two common options. The time-domain algorithms, often referred to as backprojection (BP) class algorithms [19], are most accurate and can easily adapt to all SAR configurations. Due to their computational complexity and the poor ability to integrate accurate autofocus algorithm into its imaging process, their use is restricted. The $\omega - k$ algorithm is an ideal solution without approximation in range cell migration (RCM) correction, which can focus data up to very high-resolution values regardless of their azimuth and range bandwidth. However, it is only applicable to spaceborne SAR data with a straight sensor trajectory and can only perform the range-independent motion compensation (MoCo). In addition, the Stolt interpolation makes it to be time-consuming . The extended Omega-K algorithm (EOKA) [20,21] is proposed to integrate the high precise range-dependent MoCo but it is still inefficient due to the Stolt interpolation.

For efficiency reasons, the chirp scaling class algorithms are still attractive. Efforts have been made to modify the CSA to process the low frequency SAR data. Without the interpolation, the chirp scaling class algorithms are effective and phase-preserving. The nonlinear chirp scaling algorithm (NCSA) [13] is proposed to take into account the cubic range-independent coupling phase and the range dependence of secondary range compression (SRC) term, which has better performance than CSA on processing the raw data of highly squint or low frequency SAR cases. Whereas the range dependence of cubic- and higher order terms are neglected. Some modified NCSAs are proposed in References [22–24], which resolve the high-order range-independent coupling terms. However, the cubic and higher order range-dependent coupling terms are still neglected. Besides, the first order approximation of range frequency modulation (FM) rate will introduce a quadratic phase error in the spectrum and degrades the focusing quality of image. Thus, the range focusing depth is restricted, which shows that the NCSA may not be suitable for the high-resolution low frequency SAR processing. A helpful comparison of the BP class algorithm, $\omega - k$ algorithm, EOKA and NCSA can be found in References [22,25]. In References [26,27], a generalized chirp scaling algorithm (GCSA) is developed for the SAR systems operating on wide bandwidths at low frequencies. The GCSA is an efficient arbitrary-order CSA that processes the data using the appropriate number of the approximation terms. Both the higher range-independent terms and range-dependent terms are considered. The GCSA efficiently extends the utility of frequency domain processing for high-resolution low frequency SAR systems. However, the imaging quality of GCSA also decreases as the fractional bandwidth get larger and beamwidth gets wider. In addition, the improvement of focus quality is not significant when the order is greater than 3rd and the edge targets of the range swath still have obvious degradation. Two main reasons lead to this phenomenon. One is that the residual coupling terms are still significant and the other is the error caused by the linear approximation of stationary phase point when solving

the fast Fourier transform (FFT) expression. The linear approximation makes the range-dependent high-order phase terms not effectively compensated, even if a higher-order model is used.

The aim of this study is to overcome these two limits of GCSA and propose a more accurate approach than the GCSA for processing wide-swath, high-resolution low frequency SAR data. To our knowledge, the Lagrange inversion [28–31] gives the power series representation of the inverse of an analytic function, which is quite suitable for calculating the expression of stationary phase point. This paper utilizes Lagrange inversion to calculate a more precise expression of stationary phase point, while compensating for all range-independent coupling terms above 3rd order. In our approach, the high-order chirp scaling technique is extended to achieve the effect of the range-variant filtering required in high order phase terms. The experimental results show that the improved algorithm has a better focusing performance than the original GCSA. The resolution, sidelobe level and range focusing depth are significantly improved.

This paper is organized as follows. In Section 2, the signal model of high-resolution low frequency SAR is analyzed and the limitations of existing GCSA are briefly described. Then, a principle to determine the required order of range-dependent coupling phase is presented. In Section 3, an improved GCSA based on the Lagrange inversion theorem is introduced to focus the high-resolution low frequency SAR data. Focused results obtained by the conventional GCSA and the improved GCSA are presented and analyzed to verify our analysis in Section 4. A discussion is givne in Section 5. Finally, conclusions are drawn in Section 6.

## 2. Background and Problem Statement

### 2.1. Signal Model

The 2D spectrum is a key for the frequency domain algorithm development. In our analysis, we only consider the phase terms of the SAR signal and ignore the initial phase. The range-dependent SAR transfer function in the 2D frequency domain can be expressed as [14]

$$\Phi\left(f_\tau, f_\eta; R_0\right) = -\frac{4\pi R_0 f_0}{c}\sqrt{D^2(f_\eta) + \frac{2f_\tau}{f_0} + \frac{f_\tau^2}{f_0^2}} - \frac{\pi f_\tau^2}{K_r} \tag{1}$$

where $f_\tau$ and $f_\eta$ represent the range frequency and the azimuth frequency, respectively. $R_0$ is the closest slant range from a point target to the radar trajectory, $f_0$ is the carrier frequency, $c$ is the speed of light, $K_r$ is the range chirp rate, $D(f_\eta) = \sqrt{1 - \frac{c^2 f_\eta^2}{4V_r^2 f_0^2}}$ is the migration factor, $V_r$ is the moving velocity of the radar platform.

The first term in Equation (1) represents the coupling relationship between the range frequency and the azimuth frequency, which is called the range-azimuth coupling term. It varies with the slant range. The second term is the range modulation term. The square root term can be expanded into a Taylor series with respect to $f_\tau$ and kept up to the $n$th order,

$$p\left(f_\tau, f_\eta; n\right) = D(f_\eta) + \frac{f_\tau}{f_0 D(f_\eta)} + \frac{D^2(f_\eta) - 1}{2f_0^2 D^3(f_\eta)}f_\tau^2 + \sum_{i=3}^{n}\gamma_i f_\tau^i \tag{2}$$

where $\gamma_i$ denotes the coefficient of the $i$th term and is given in Appendix A. The first term in Equation (2) corresponds to the azimuth modulation. The second term corresponds to the RCM (first-order coupling term). The third term denotes the SRC (second-order coupling term). The remainder higher order terms donates the high-order range-azimuth coupling. Different frequency algorithms are based on specific order approximations of this equation. For example, the CSA uses a 2nd order model and the NCSA uses a 3rd moder model. For the low frequency SAR with a small bandwidth and narrow beamwidth, quadratic approximation is enough, whereas for large bandwidth the high-order terms become significant.

Reconsider the phase in Equation (1). Actually, the coupling phase term can be decomposed into two parts: the range-independent terms and the range-dependent terms, namely,

$$\Phi\left(f_\tau, f_\eta; R_0, n\right) = -\frac{4\pi R_c f_0}{c} p\left(f_\tau, f_\eta; n\right) - \frac{4\pi \Delta R f_0}{c} p\left(f_\tau, f_\eta; n\right) - \frac{\pi f_\tau^2}{K_r} \tag{3}$$

where $R_c$ represents the slant range of scene center, $\Delta R = R_0 - R_c$ represents the difference in slant range between the point target and the scene center point. The first term donates the range-independent coupling phase and the second term donates the range-dependent coupling phase, which varies with the slant range $R_0$.

### 2.2. The Limitations of the Conventional GCSA

The phase error due to the Taylor approximation can be expressed as

$$\Phi_{error}\left(f_\tau, f_\eta; R_0, n\right) = -\frac{4\pi R_0 f_0}{c}\left(\sqrt{D^2(f_\eta) + \frac{2f_\tau}{f_0} + \frac{f_\tau^2}{f_0^2}} - p(f_\tau, f_\eta; n)\right) \tag{4}$$

Note that this error gets larger when the slant range $R_0$ increases, the maximum range frequency $f_{\tau,\max}$ increases, the maximum azimuth frequency $f_{\eta,\max}$ increases, while the center frequency $f_0$ decreases. In high-resolution, low frequency and far range situations, the approximation error is large and a high order model is required.

To illustrate the phase error of Taylor expansion, a numerical analysis is carried out. Assume that the center frequency $f_0$ equals to 600 MHz, bandwidth $B_r$ is 300 MHz, beamwidth is 29° and the target slant range $R_0$ is 12 km. Figure 1 shows the relationship between the phase errors and the range frequency for different order approximations. The maximal phase error of 4th order model is 1025° and the maximal phase error of 6th order approximation is about 81.48°, which indicates that the coupling phase terms above 6th order still have an important influence on the image quality. High-resolution imaging methods should take these terms into account.

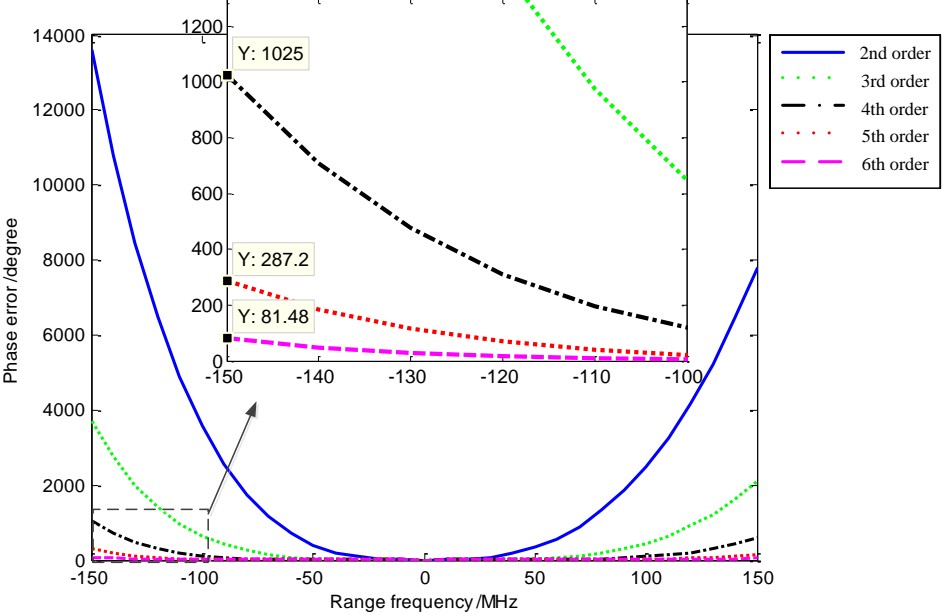

**Figure 1.** Phase errors of 2nd to 6th order Taylor series approximation. The point target at a range of 12 km with a center frequency of 600 MHz, a bandwidth of 300 MHz and a beamwidth of 29°.

In Reference [26], Zaugg et al. gives a guideline to determine required order $n$th from Equation (4) for proper focusing: If less than 30% of the support band has a phase error greater than $\pi/10$, then one

can predict less than a 20% loss in the azimuth resolution defocusing. The conventional GCSA first compensates the 3rd to $n$th order range-independent coupling terms and then uses the high-order chirp scaling filter to compensate the range-dependent coupling phase. The coupling phase terms above $n$th order are neglected.

In the case of low-frequency high-resolution SAR with large bandwidth and wide beam, the coupling phase above 6th order still has a large proportion. Therefore, a higher order model is needed. However, when we use a model higher than 3rd order, the linear approximation of the stationary phase point when using the principle of stationary phase (POSP) will bring serious errors, which severely degrades the actual performance of high-order models in conventional GCSA. This error makes the high-order range-dependent coupling phase not to be accurately compensated, even though a higher order model is used. This is because the nonlinear FM component of signal cannot be neglected. Besides, the complexity of algorithm design will increase when using a higher order model.

Using the same parameters presented in the previous analysis, Figure 2 shows the residual range migrations of a point target at a range of 12 km after processing by the conventional GCSA. It is assumed here that the reference slant range is 10 km. The range migration crosses several range gates even using a 7th order model. This is because the conventional algorithm does not accurately compensate the range-dependent coupling terms.

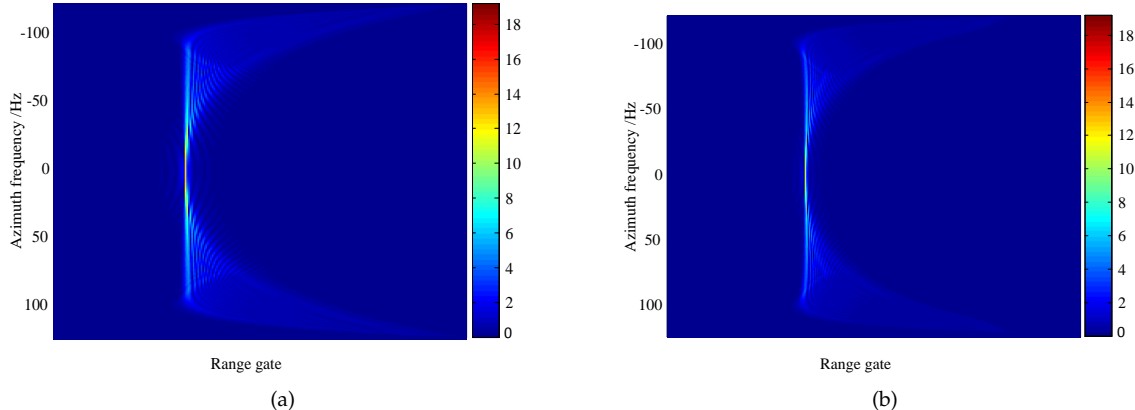

(a)                                                    (b)

**Figure 2.** Range migrations of point target at a range of 12 km after RCMC. (**a**) Algorithm in Reference [26] (6th order model). (**b**) Algorithm in Reference [26] (7th order model).

### 2.3. New Principle to Determine the Required Order of Range Frequency

According to the previous analysis, the phase error of 6th order model is still significant. In fact, the range-independent couplings can be compensated by the reference range phase. Figure 3 shows the range-dependent coupling phase errors of different order models. If the range-independent coupling terms are firstly compensated, the maximum range-dependent phase error of 6th order model is 13.58°, which has a small effect on the imaging results.

Therefore, if the range-independent coupling terms are firstly compensated, we should only consider the range-dependent coupling terms to determine the required order. Thus, a new principle of the proposed method to determine the required order is given here. Assume that the $M$th order range-dependent coupling term should be taken into consideration, the phase error should meet

$$-\frac{4\pi\Delta R f_0}{c}\left(\sqrt{D^2(f_\eta) + \frac{2f_\tau}{f_0} + \frac{f_\tau^2}{f_0^2}} - p\left(f_\tau, f_\eta; M\right)\right)\Bigg|_{f_\tau=\frac{B_r}{2}, f_\eta=\frac{2f_0 V_r}{c}\sin\theta} \leq \delta\pi \qquad (5)$$

where $\theta$ is the beamwidth, $\delta$ equals to 1/10 in this paper and it can be set to a larger value when the imaging accuracy is not high.

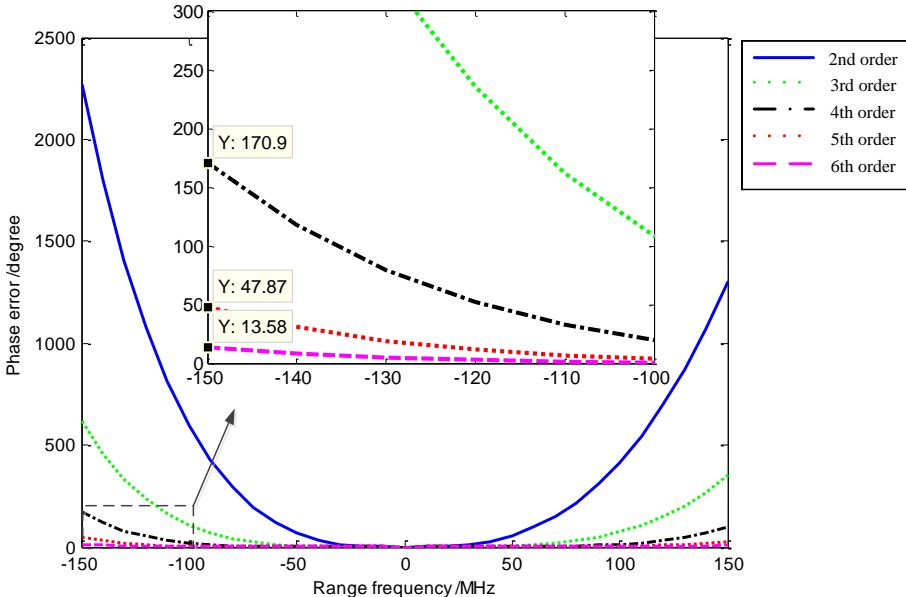

**Figure 3.** Range-dependent coupling phase errors of 2nd to 6th order Taylor series approximation. The point target at the slant range 12 km with a center frequency of 600 MHz, a bandwidth of 300 MHz, a beamwidth of 29° and the reference slant range of 10 km.

Inequality (5) is an important basis for the proposed algorithm. It can be seen that the phase error is positively correlated with the range bandwidth $B_r$, the beamwidth $\theta$ and the scene size $2\Delta R$ and negatively correlated with the center frequency $f_0$. If the center frequency and scene size are given, the required order is determined by the range bandwidth and the azimuth beamwidth.

Figure 4 shows an example of a P-band SAR system. The center frequency is 800 MHz and the scene size equals to 600 m. The required order varies with different beamwidth and fractional bandwidth. As can be seen from Figure 4, under the given parameters, the 2nd order model can only process data with a fractional bandwidth of less than 0.34. As the beamwidth increases, the fractional bandwidth value that can be processed is smaller. For low frequency SAR systems with large bandwidths and wide beams, a higher order range-dependent coupling phase should be considered.

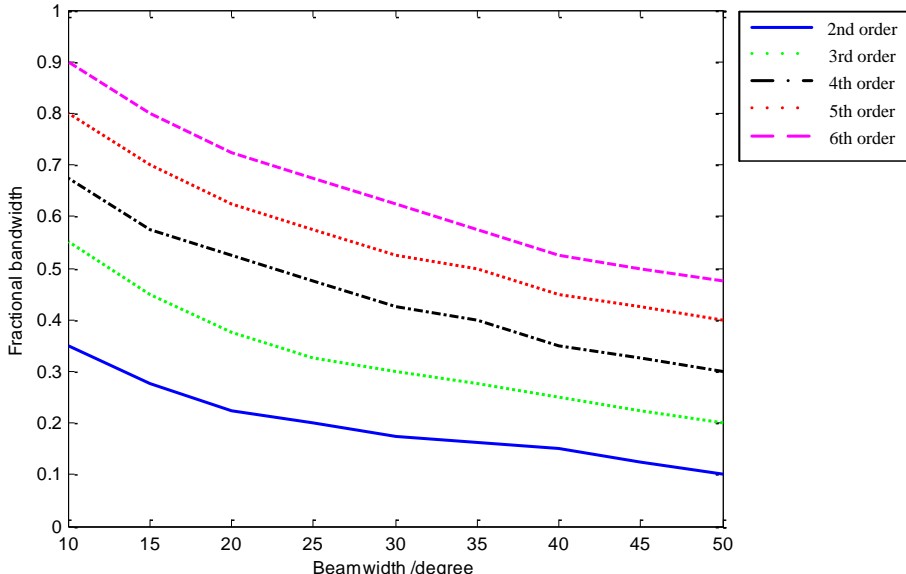

**Figure 4.** An example of the required order of different beamwidth and fractional bandwidth in a P-band synthetic aperture radar (SAR) system.

### 3. The Improved GCSA Based on Lagrange Inversion Theorem

*3.1. Procedure of Algorithm*

Based on the analysis in Section 2, we proposed an improved GCSA in this section. The flow chart of the proposed algorithm is shown in the Figure 5. There are two main improvements to the algorithm. The first is to perform all the high-order ($\geq$3rd) reference phase compensation in the 2D frequency domain. The second is to use the Lagrange inversion theorem to solve the expression of the range FFT and inverse FFT (IFFT). The derivation of the proposed algorithm is given in Section 3.2. The steps of the proposed algorithm are as follows:

Step 1: Select the parameter $M$ according to the principle of Equation (5). Calculate $q_2 \sim q_M$, $X_3 \sim X_M$ and $C_2 \sim C_M$ according to Equations (25) and (26). Then, 2D FFT is implemented to transfer the data into 2D frequency domain.

Step 2: Multiply the range-independent high-order phase correction (HOPC) and perturbation equations in the 2D frequency domain. And a range IFFT is carried out to transfer the data into the range-Doppler (RD) domain.

Step 3: Multiply the high-order chirp scaling phase function. Then, the data are transformed into 2D frequency domain by range FFT.

Step 4: Multiply the range compression function to perform the RC, SRC and bulk RCMC. Then, the data are transferred into RD domain by IFFT along the range.

Step 5: Multiply the azimuth compression and residual phase correction function. And the images can be obtained by azimuth IFFT.

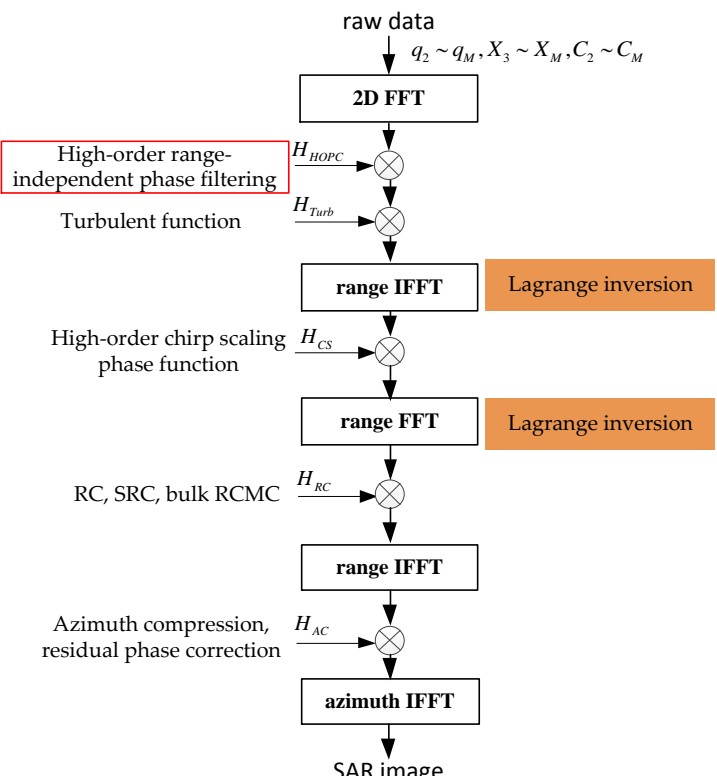

**Figure 5.** The flow chart of the proposed algorithm.

*3.2. Theoretical Formulation*

As mentioned in Section 2, the high-order terms of high-resolution low frequency SAR account for a large proportion. If it can not be eliminated, the image will be deteriorated dramatically. In order to reduce the phase error and improve imaging accuracy, the range-independent high-order ($\geq$3rd)

coupling phases of Equation (3) is firstly compensated in the 2D frequency domain. The 3rd order is chosen because of the need to preserve the chirp information of the range signals. The HOPC function at the reference range is given by

$$H_{HOPC} = \exp\left[\frac{4\pi R_c f_0}{c}\left(\sqrt{D^2(f_\eta) + \frac{2f_\tau}{f_0} + \frac{f_\tau^2}{f_0^2}} - D(f_\eta) - \frac{f_\tau}{f_0 D(f_\eta)} - \frac{D^2(f_\eta) - 1}{2f_0^2 D^3(f_\eta)} f_\tau^2\right)\right] \quad (6)$$

Suppose that the *M*th range-dependent coupling should be taken into account. The residual signal after HOPC can be expressed as

$$\Phi(f_\tau, f_\eta; R_0) = -\frac{4\pi R_0 f_0}{c} D(f_\eta) - \frac{4\pi R_0}{cD(f_\eta)} f_\tau - \frac{\pi}{K_m} f_\tau^2 - \frac{4\pi \Delta R f_0}{c} \sum_{i=3}^{M} \gamma_i f_\tau^i \quad (7)$$

where $K_m$ donates the range FM rate and can be expressed as

$$K_m = \frac{K_r}{1 - K_r \dfrac{cR_0 f_\eta^2}{2V_r^2 f_0^3 D(f_\eta)^3}} \quad (8)$$

It can be found that the residual high-order coupling becomes zero at the reference range but the residual high-order phase at other ranges increases as the target away from the reference range. Then we filter the data with a turbulent function

$$H_{Turb} = \exp\left(j\pi \sum_{i=3}^{M} X_i f_\tau^i\right) \quad (9)$$

This filtering step provides an accurate accommodation of the range dependence of the high-order terms. After the turbulent compensation, the phase can be expressed as

$$\Phi_1(f_\tau, f_\eta; R_0) = -\frac{4\pi R_0 f_0}{c} D(f_\eta) - \frac{4\pi R_0}{cD(f_\eta)} f_\tau - \frac{\pi}{K_m} f_\tau^2 + \pi \sum_{i=3}^{M} \left(X_i - \frac{4\pi \Delta R f_0}{c} \gamma_i\right) f_\tau^i$$
$$= \phi_0 + \phi_1 f_\tau + \phi_2 f_\tau^2 + \sum_{i=3}^{M} \phi_i f_\tau^i \quad (10)$$

with

$$\begin{aligned}
\phi_0 &= -\frac{4\pi R_0 f_0}{c} D(f_\eta) \\
\phi_1 &= -\frac{4\pi R_0}{cD(f_\eta)} = -2\pi \tau_d \\
\phi_2 &= -\frac{\pi}{K_m} \\
\phi_i &= \pi\left[X_i - 2f_0 D(f_\eta)\gamma_i \Delta\tau\right] (3 \le i \le M)
\end{aligned} \quad (11)$$

Then the range IFFT is performed along the range direction. Based on the POSP, the relationship between $\tau$ and $f_\tau$ can be expressed as

$$\tau = -\frac{1}{2\pi}(\phi_1 + 2\phi_2 f_\tau + ... + M\phi_M f_\tau^{M-1}) \quad (12)$$

In Reference [26], only the first order term of $f_\tau$ is retained when solving the stationary phase point, that is, $f_\tau = -\frac{\pi}{\phi_2}\left(\tau + \frac{\phi_1}{2\pi}\right)$. This approximation is effective in most cases. However, as the center frequency decreases, the beamwidth and bandwidth increase, this approximation introduces severe phase errors, resulting in significant degradation throughout the image. To solve this problem, we use the Lagrange inversion to find a more accurate solution for $f_\tau$.

In mathematical analysis, the Lagrange inversion theorem gives the Taylor series expansion of the inverse function of an analytic function and it can be expressed as Theorem 1 [28–31].

**Theorem 1.** *Suppose w is defined as a function of z by an equation of the form $w = h(z)$, where h is analytic at a point $z_0$ and $h'(z_0) \neq 0$. Then it is possible to invert or solve the equation for z, expressing it in the form $z = g(w)$ given by a power series*

$$g(w) = z_0 + \sum_{n=1}^{\infty} g_n (w - h(z_0))^n \tag{13}$$

*where*

$$g_n = \frac{1}{n!} \lim_{z \to z_0} \frac{d^{n-1}}{dz^{n-1}} \left[ \left( \frac{z - z_0}{h(z) - h(z_0)} \right)^n \right] \tag{14}$$

*$g(w)$ represents an analytic function of w in a neighbourhood of $w = h(z_0)$.*

As is shown in Equation (12), $\tau$ is a power series of $f_\tau$, that is, $\tau = h(f_\tau)$. Let $z_0 = 0$ and $h(z_0) = -\frac{\phi_1}{2\pi}$. According to the Lagrange inversion theorem, the stationary phase point can be given by

$$
\begin{aligned}
f_\tau = & -\frac{\pi}{\phi_2} \left( \tau + \frac{\phi_1}{2\pi} \right) - \frac{3\pi^2 \phi_3}{2\phi_2^3} \left( \tau + \frac{\phi_1}{2\pi} \right)^2 + \frac{\pi^3 \left( -9\phi_3^2 + 4\phi_2\phi_4 \right)}{2\phi_2^5} \left( \tau + \frac{\phi_1}{2\pi} \right)^3 \\
& -\frac{5\pi^4 \left( 27\phi_3^3 - 24\phi_2\phi_3\phi_4 + 4\phi_2^2\phi_5 \right)}{8\phi_2^7} \left( \tau + \frac{\phi_1}{2\pi} \right)^4 + \frac{3\pi^5 \left( -189\phi_3^4 + 252\phi_2\phi_3^2\phi_4 - 60\phi_2^2\phi_3\phi_5 + 8\phi_2^2 \left( -4\phi_4^2 + \phi_2\phi_6 \right) \right)}{8\phi_2^9} \left( \tau + \frac{\phi_1}{2\pi} \right)^5 + \dots
\end{aligned} \tag{15}
$$

The detailed derivation of Equation (15) is given in Appendix C. Therefore, the IFFT expression of Equation (10) can be expressed as

$$\Phi_2(\tau, f_\eta; R_0) = \phi_0 + A_2(\tau - \tau_d)^2 + A_3(\tau - \tau_d)^3 + \dots + A_M(\tau - \tau_d)^M \tag{16}$$

with (Here, only $A_2 \sim A_6$ are given due to space restraints.)

$$
\begin{cases}
A_2 = -\frac{\pi^2}{\phi_2} \\
A_3 = -\frac{\pi^3 \phi_3}{\phi_2^3} \\
A_4 = \frac{\pi^4 \left( -9\phi_3^2 + 4\phi_2\phi_4 \right)}{4\phi_2^5} \\
A_5 = -\frac{\pi^5 \left( 27\phi_3^3 - 24\phi_2\phi_3\phi_4 + 4\phi_2^2\phi_5 \right)}{4\phi_2^7} \\
A_6 = \frac{\pi^6 \left( -189\phi_3^4 + 252\phi_2\phi_3^2\phi_4 - 60\phi_2^2\phi_3\phi_5 + 8\phi_2^2 \left( -4\phi_4^2 + \phi_2\phi_6 \right) \right)}{8\phi_2^9} \\
\dots
\end{cases} \tag{17}
$$

where $\tau_d = h(z_0) = 2R_0 / (cD(f_\eta))$ is the time delay in RD domain. The variation of $\tau_d$ with $f_\eta$ is called range migration, which must be removed before azimuth compression. The shape of the range migration trajectory depends on the target slant $R_0$.

The high-order CS function can be given by

$$H_{CS}(\tau, f_\eta) = \exp \left[ j\pi q_2 \left( \tau - \tau_{ref} \right)^2 + j\pi \sum_{i=3}^{M} q_i \left( \tau - \tau_{ref} \right)^i \right] \tag{18}$$

where $\tau_{ref} = 2R_c / (cD(f_\eta))$ is the reference trajectory. This step aims to compensate the range-dependent coupling caused by the large fractional bandwidth and wide beamwidth. The phase after high-order CS filter can be expressed as

$$\Phi_3(\tau, f_\eta; R_0) = \phi_0 + A_2(\tau - \tau_d)^2 + \sum_{i=3}^{M} A_i(\tau - \tau_d)^i + \pi q_2 \left( \tau - \tau_{ref} \right)^2 + \pi \sum_{i=3}^{M} q_i \left( \tau - \tau_{ref} \right)^i \tag{19}$$

According to the chirp scaling principle, the desired trajectory $\tau_s$ of the target located at range $R_0$ has the same shape as the reference trajectory and the relationship between $\tau_s$, $\tau_{ref}$ and $\tau_d$ can be expressed as

$$
\begin{aligned}
\tau_{ref} &= \tau_s - \alpha \Delta\tau \\
\tau_d &= \tau_s - (\alpha - 1)\Delta\tau
\end{aligned}
\tag{20}
$$

where $\Delta\tau = 2\Delta R / (cD(f_\eta))$ and $\alpha = D(f_\eta)/D(f_{\eta c})$. Using the relationship in Equation (20) and expand Equation (19) at $\tau_s$, we obtain

$$
\Phi_4(\tau, f_\eta; R_0) = \phi_0 + \pi C_0(\Delta\tau) + \pi \sum_{i=1}^{M} C_i(\tau - \tau_s)^i
\tag{21}
$$

The expressions for coefficients $C_i$ are given in Appendix B. As can be seen form Equation (11), parameters $A_i$ and $C_i$ imply the range FM rate $K_m$, which is dependent on azimuth frequency and slant range. Imaging performance is affected by approximations to $K_m$. To model the range range-dependence of $K_m$, we expand it at the reference slant range with Taylor series and keep up to the second order,

$$
K_{m,app} \approx K_f + K_s K_f^2 \Delta\tau + K_s^2 K_f^3 \Delta\tau^2
\tag{22}
$$

where $K_f$ represents the FM rate at the reference slant range and can be expressed as

$$
K_f = \frac{K_r}{1 - K_r \tau_{ref} K_s}
\tag{23}
$$

with

$$
K_s = \frac{c^2 f_\eta^2}{4v^2 f_0^3 D(f_\eta)^2}
\tag{24}
$$

According to Appendix B, the expressions of $q_i(i \geq 2)$ and $X_i(i \geq 3)$ can be expressed as

$$
\begin{cases}
q_2 = K_f \frac{1-\alpha}{\alpha} \\
q_3 = \frac{K_f^2 K_s(1-\alpha)}{3\alpha} \\
X_3 = \frac{K_s(\alpha-2)}{3K_f(\alpha-1)} \\
q_4 = -\frac{K_f^3 \left(9K_f K_s(\alpha-1)X_3 + 2Ks^2 - 6D(f_\eta)f_0(\alpha-1)\gamma_3\right)}{12\alpha} \\
X_4 = \frac{9K_f K_s(\alpha-2)X_3 - 27K_f^2(\alpha-1)X_3^2 + 2Ks^2 - 6D(f_\eta)f_0(\alpha-1)\gamma_3}{12K_f(\alpha-1)} \\
q_5 = -\frac{45K_f^6 K_s(\alpha-1)X_3^2 + 12K_f^5 X_3\left(K_s^2 - 3D(f_\eta)f_0(\alpha-1)\gamma_3\right)}{20\alpha} \\
\quad - \frac{16K_f^5 K_s(\alpha-1)X_4 - 4K_f^4 D(f_\eta)f_0(3K_s\gamma_3 + 2(\alpha-1)\gamma_4)}{20\alpha} \\
X_5 = -\frac{-45K_f^2 K_s(\alpha-2)X_3^2 + 12K_f X_3\left(-K_s^2 + 10K_f(\alpha-1)X_4 + 3D(f_\eta)f_0(\alpha-2)\gamma_3\right)}{20K_f(\alpha-1)} \\
\quad - \frac{4D(f_\eta)f_0(3K_s\gamma_3 + 2(\alpha-2)\gamma_4) - 16K_f K_s(\alpha-2)X_4 + 135K_f^3(\alpha-1)X_3^3}{20K_f(\alpha-1)} \\
\dots
\end{cases}
\tag{25}
$$

And coefficients $C_i$ becomes

$$
\begin{cases}
C_1 = 0 \\
C_2 = q_2 + K_f \\
C_3 = q_3 + K_f^3 X_3 \\
C_4 = q_4 + \frac{9}{4}K_f^5 X_3^2 + K_f^4 X_4 \\
C_5 = q_5 + \frac{27}{4}K_f^7 X_3^3 + 6K_f^6 X_3 X_4 + K_f^5 X_5 \\
\dots
\end{cases}
\tag{26}
$$

After the chirp scaling operation, a range FFT is carried to transform Equation (21) into the 2D frequency domain. Similarly, $f_\tau$ is a power series of $\tau$,

$$f_\tau = \frac{1}{2}\left(2C_2(\tau - \tau_s) + 3C_3(\tau - \tau_s)^2 + ...MC_M(\tau - \tau_s)^{M-1}\right) \tag{27}$$

Use the Lagrange inversion theorem again. Let $z_0 = \tau_s$ and $h(z_0) = 0$, the stationary phase point is given by

$$\tau = \tau_s + \frac{1}{C_2}f_\tau - \frac{3C_3}{2C_2^3}f_\tau^2 + \frac{\left(\frac{9C_3^2}{2} - 2C_2C_4\right)}{C_2^5}f_\tau^3 - \frac{\left(\frac{135C_3^3}{8} - 15C_2C_3C_4 + \frac{5}{2}C_2^2C_5\right)}{C_2^7}f_\tau^4$$
$$+ \frac{\left(\frac{567C_3^4}{8} - \frac{189}{2}C_2C_3^2C_4 + 12C_2^2C_4^2 + \frac{45}{2}C_2^2C_3C_5 - 3C_2^3C_6\right)}{C_2^9}f_\tau^5 + ... \tag{28}$$

Thus, the phase after range FFT can be expressed as

$$\Phi_5(f_\tau, f_\eta) = \phi_0 + \pi C_0(\Delta\tau) - 2\alpha\tau_d f_\tau + \pi\sum_{i=1}^{M} E_i f_\tau^i \tag{29}$$

with

$$\begin{cases} E_1 = -2\tau_{ref}(1 - \alpha) \\ E_2 = -\frac{1}{C_2} \\ E_3 = \frac{C_3}{C_2^3} \\ E_4 = \frac{\left(-9C_3^2 + 4C_2C_4\right)}{4C_2^5} \\ E_5 = \frac{\left(27C_3^3 - 24C_2C_3C_4 + 4C_2^2C_5\right)}{4C_2^7} \\ ... \end{cases} \tag{30}$$

The first term of Equation (29) represents the azimuth compression phase, the second term is the residual phase, the third term is the linear phase corresponding to the target position, the remainder are related to the range compression (RC), SRC and bulk RCM, which are range invariant. Thus, the bulk RCM, SRC and RC can be compensated by a conjugate multiply in the 2D frequency domain. The filtering function can be expressed as

$$H_{RC}(f_\tau, f_\eta) = \exp(-j\pi\sum_{i=1}^{M} E_i f_\tau^i) \tag{31}$$

After the compensation, the phase of signal can be expressed as

$$\Phi_6(f_\tau, f_\eta) = \phi_0 + \pi C_0(\Delta\tau) - 2\pi\alpha\tau_d f_\tau \tag{32}$$

An IFFT is carried out along the range direction and the signal phase becomes

$$\Phi_7(\tau, f_\eta) = \phi_0 + \pi C_0(\Delta\tau) \tag{33}$$

Therefore, the azimuth compression function is given by

$$H_{AC}(\tau, f_\eta) = \exp\left[j\pi\left(\frac{4\pi R_0 f_0}{c}(D(f_\eta) - 1) - C_0(\Delta\tau)\right)\right] \tag{34}$$

Finally, the echo data are transformed into 2D time domain by a azimuth IFFT and the focused image is obtained.

## 4. Experiment Results and Analysis

In this section, we provide some imaging results to demonstrate the performance of proposed algorithm and the analysis of principle. The system parameters are listed in Table 1. The parameters of platform velocity, pulse duration and center slant range of each SAR system are set to the same value. The reference slant range is selected as the scene center slant range. The theoretical azimuth and range resolutions are evaluated by $\rho_a = 0.886c/(4f_0 \sin(\theta/2))$ and $\rho_r = 0.886c/(2B_r)$. The maximum Doppler frequency can be expressed as

$$f_D = \frac{2V_r}{\lambda_{min}} \sin\left(\frac{\theta_{max}}{2}\right) \tag{35}$$

where $\lambda_{min}$ and $\theta_{max}$ are the minimum signal wavelength and the maximum azimuth angle. The PRF must not be chosen less than two times of $f_D$. The range oversampling rate is set to 1.2.

**Table 1.** SAR system simulation parameters.

| Parameters | P-band | L-band |
|---|---|---|
| Center frequency $f_0$ (MHz) | 600 | 1360 |
| Bandwidth $B_r$ (MHz) | 300 | 272/544/816/1088 |
| Fractional bandwidth (%) | 50 | 20/40/60/80 |
| Beamwidth $\theta$ (°) | 29 | 11 |
| Azimuth resolution $\rho_a$ (m) | 0.44 | 0.5 |
| PRF (Hz) | 240 | 240 |
| Velocity of platform $V_r$ (m/s) | 100 | 100 |
| Pulse duration $T_r$ (us) | 10 | 10 |
| Center slant range $R_c$ (km) | 10 | 10 |

Firstly, to investigate the effects of two improvements in the proposed algorithm, a P-band SAR with a center frequency of 600 MHz was simulated. The fractional bandwidth is set to 50% (corresponding to a range resolution of 0.44 m) and the beamwidth is set to 29° (corresponding to an azimuth resolution of 0.44 m). Nine targets are arranged in the illuminated scene along the azimuth center at different distances form the reference range with an interval of 200 m. The conventional GCSA in Reference [26] is used in comparison with the proposed algorithm. According to the principle in Section 2.3, the data is processed with a 6th order model. To highlight the effect of Lagrange inversion, the conventional GCSA with Lagrange inversion (ignoring the range-independent coupling terms above 6th order) is also used in comparison. The 2D focused images of targets at the ranges $R_c$, $R_c + 800$ m and $R_c + 1600$ m are shown in Figure 6. In these figures, contour maps donate the 2D focusing quality. To evaluate the quality of these images quantitatively, the resolution (Res), peak sidelobe ratio (PSLR) and integrated sidelobe ratio (ISLR) along the range and azimuth directions are presented in Table 2. No wighting function or sidelobe control approach is used to obtain a fair comparison. Note that the ideal response is not completely symmetric in the range direction due to the significant range-azimuth coupling.

Figure 6a–c and Table 2 illustrate that the conventional GCSA causes deterioration of the images. The images of three targets are defocused in two directions, even at the scene center targets. The Res, PSLR and ISLR degrade considerably. This issue becomes increasingly serious as the distance increases. This problem is caused by the residual high-order range-independent coupling terms and the phase error introduced by approximate of stationary phase point. As the fractional bandwidth increases, even the scene center point has a severe degradation. Especially, the asymmetric sidelobe in range dimension is obvious. Figure 6d–f shows the imaging results obtained by the GCSA with Lagrange inversion. It can be seen that all three targets are effectively focused, which indicates that the Lagrange inversion can eliminate the range-dependent coupling terms. However, since the range-independent coupling terms above 6th order are neglected, the sidelobes in range direction are severe and the images have a certain quality degradation. The imaging results obtained by the proposed algorithm

are shown in Figure 6g–i, with better quality than Figure 6d–f. From the quality indices presented in Table 2, the imaging coherency is good over the entire swath.     These benefits stem from the compensation of coupling terms above 6th order and Lagrange inversion.

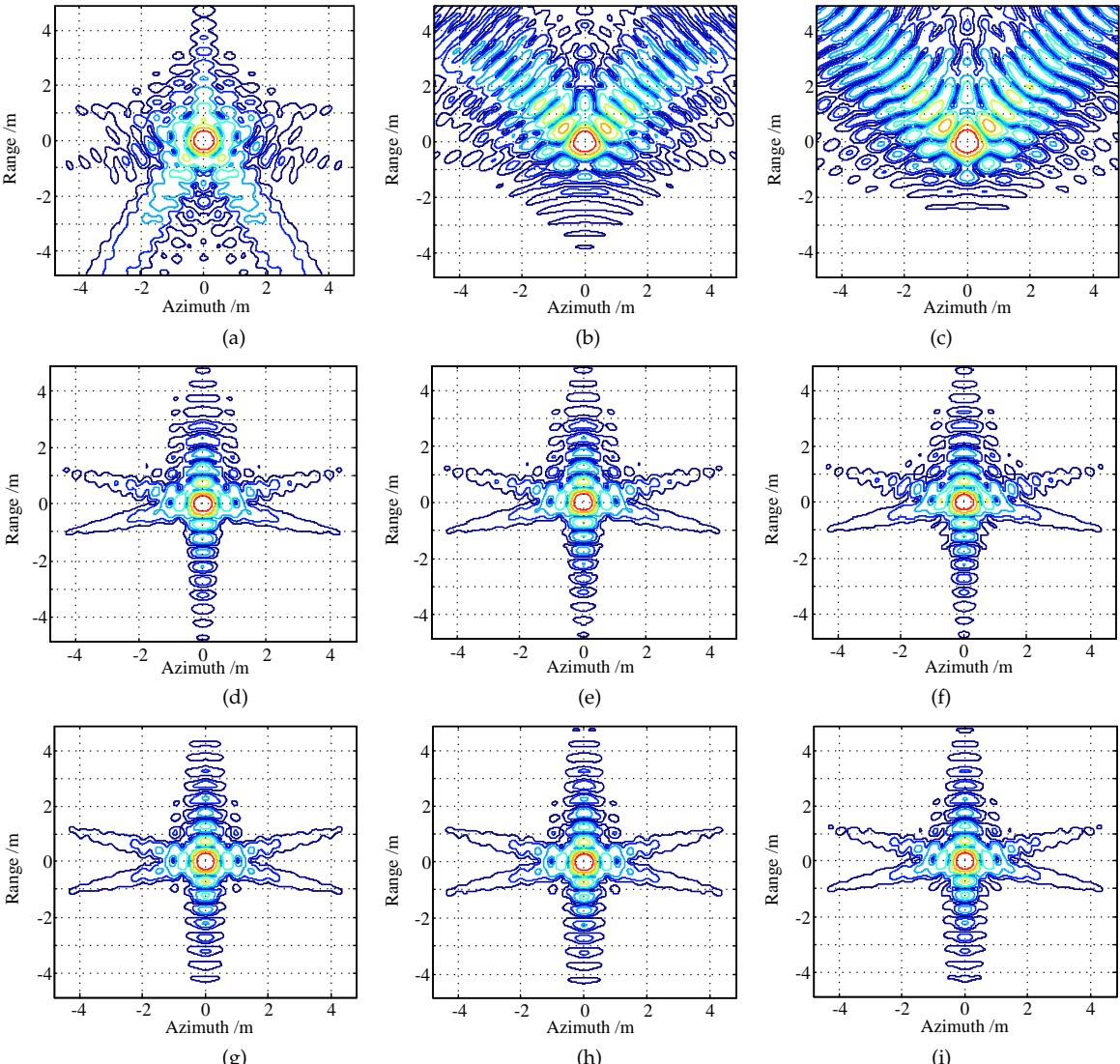

**Figure 6.** Focused results by different algorithm for P-band SAR data with a center frequency of 600 MHz. (**a**–**c**) Conventional generalized chirp scaling algorithm (GCSA). (**d**–**f**) Conventional GCSA with Lagrange inversion. (**g**–**i**) Proposed algorithm. The three subgraphs of each row correspond to contours of the targets located at $R_c$, $R_c + 800$ m and $R_c + 1600$ m, respectively. The dynamic range of contour is $-35$ dB$\sim$0 dB.

To evaluate the accuracy of proposed algorithm better, we measure the Res and differential resolutions (DRES) [32] at different slant range. Figure 7 shows the Res and DRES of nine point targets processed in different algorithms. The references for the DRES measurements are the range and azimuth resolutions obtained by $\omega - k$ algorithm. It can seen that the range and azimuth resolutions loss of GCSA is greater than 13% compared with the $\omega - k$ algorithm but the resolutions loss of the proposed algorithm is less than 1%. We can conclude that the focusing performances of the proposed algorithm are much better than the ones of GCSA and closed to the ones of $\omega - k$ algorithm. The image quality and focusing depth in range dimension are greatly improved.

**Table 2.** Measured parameters of the imaging results for Figure 6.

| Method | $R_0 - R_c$ | Azimuth | | | Range | | |
|---|---|---|---|---|---|---|---|
| | | Res /m | PSLR /dB | ISLR /dB | Res /m | PSLR /dB | ISLR /dB |
| Conventional GCSA in Reference [26] | 0 | 0.4927 | −16.9620 | −14.2797 | 0.5221 | −14.8340 | −9.9040 |
| | 800 m | 0.5344 | −21.0146 | −15.0018 | 0.5690 | −17.1714 | −11.5202 |
| | 1600 m | 0.5615 | −20.7017 | −14.6937 | 0.6341 | −16.6871 | −10.2194 |
| Conventional GCSA+Lagrange | 0 | 0.4385 | −15.0555 | −13.7173 | 0.4505 | −12.4212 | −9.7310 |
| | 800 m | 0.4385 | −15.1025 | −13.7213 | 0.4505 | −12.4613 | −9.7596 |
| | 1600 m | 0.4427 | −14.8451 | −13.2582 | 0.4518 | −12.8161 | −10.1929 |
| Proposed algorithm | 0 | 0.4365 | −15.1755 | −13.9167 | 0.4479 | −12.9714 | −10.2222 |
| | 800 m | 0.4365 | −15.1673 | −13.9233 | 0.4479 | −13.0231 | −10.2356 |
| | 1600 m | 0.4406 | −15.0641 | −13.5990 | 0.4492 | −13.2844 | −10.5722 |

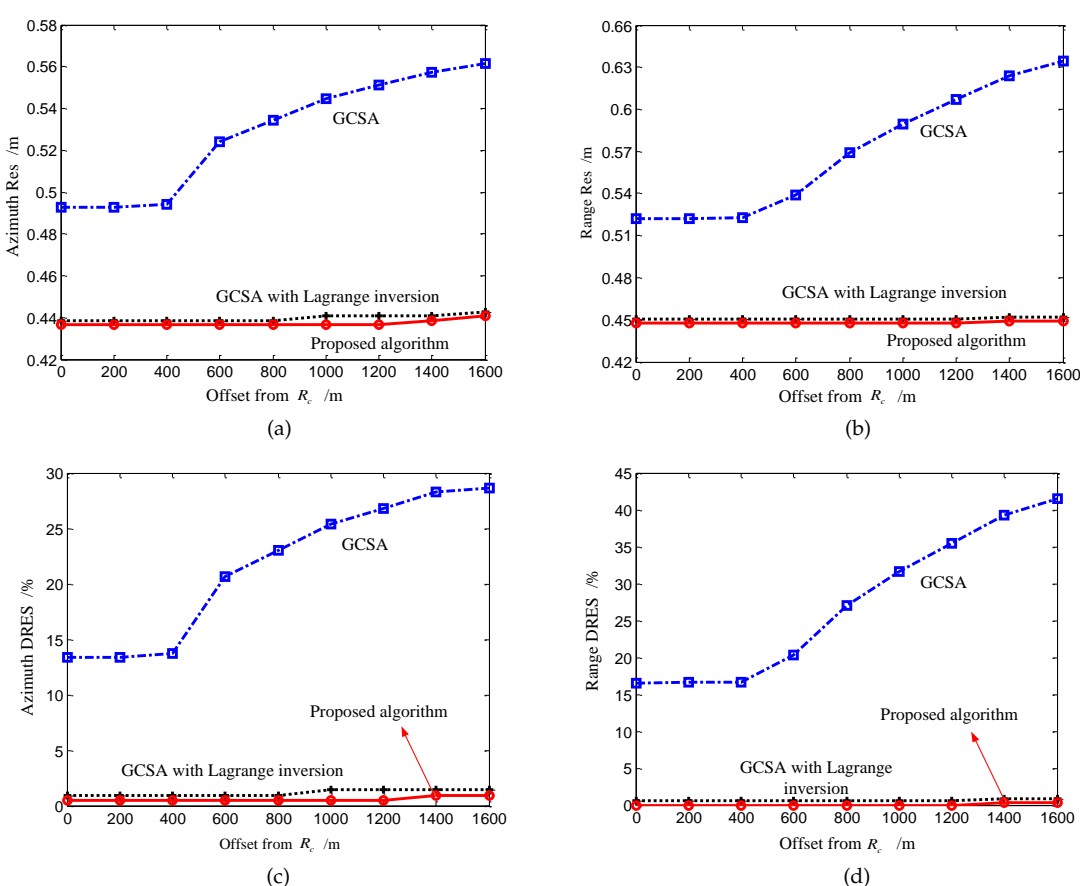

**Figure 7.** Resolutions (Res) and differential resolutions (DRES) in azimuth and range given by different algorithm, where the resolutions obtained by $\omega - k$ algorithm are references. The DRES presents the loss in spatial resolutions. Nine targets are arranged in the illuminated scene along the azimuth center at different distances form the reference range with an interval of 200 m. (**a**) Azimuth Res. (**b**) Range Res. (**c**) Azimuth DRES. (**d**) Range DRES.

Secondly, in order to better validate the performance of the proposed algorithm, the point scatterer was placed at the edge of the swath, where $R_0 - R_c$ = 2 km. A typical L-band SAR system with a center frequency of 1.36 GHz was simulated. The beamwidth is set to 11° (corresponding to an azimuth resolution of 0.5 m). The fractional bandwidth is set to 20%, 40%, 60% and 80%, respectively. According to Equation (5), four sets of data are focused by the 3rd order, 4th order, 6th order and 7th order models, respectively.

Figure 8 shows the Res and DRES versus fractional bandwidth for the GCSA and proposed algorithm. The resolutions obtained by $\omega - k$ algorithm are references. When the fractional bandwidth is 20%, both the GCSA and proposed algorithm can achieve good focusing performance. As the fractional bandwidth increases, the performance of GCSA drops dramatically. If the resolution loss threshold is 10%, the GCSA can only process the data where the fractional bandwidth is less than 30%. However, the proposed algorithm achieves nearly the theoretically resolutions for fractional bandwidths up to 80% for L-band. The resolution broadening is less than 1%, which shows the performance is greatly improved over that of the original GCSA.

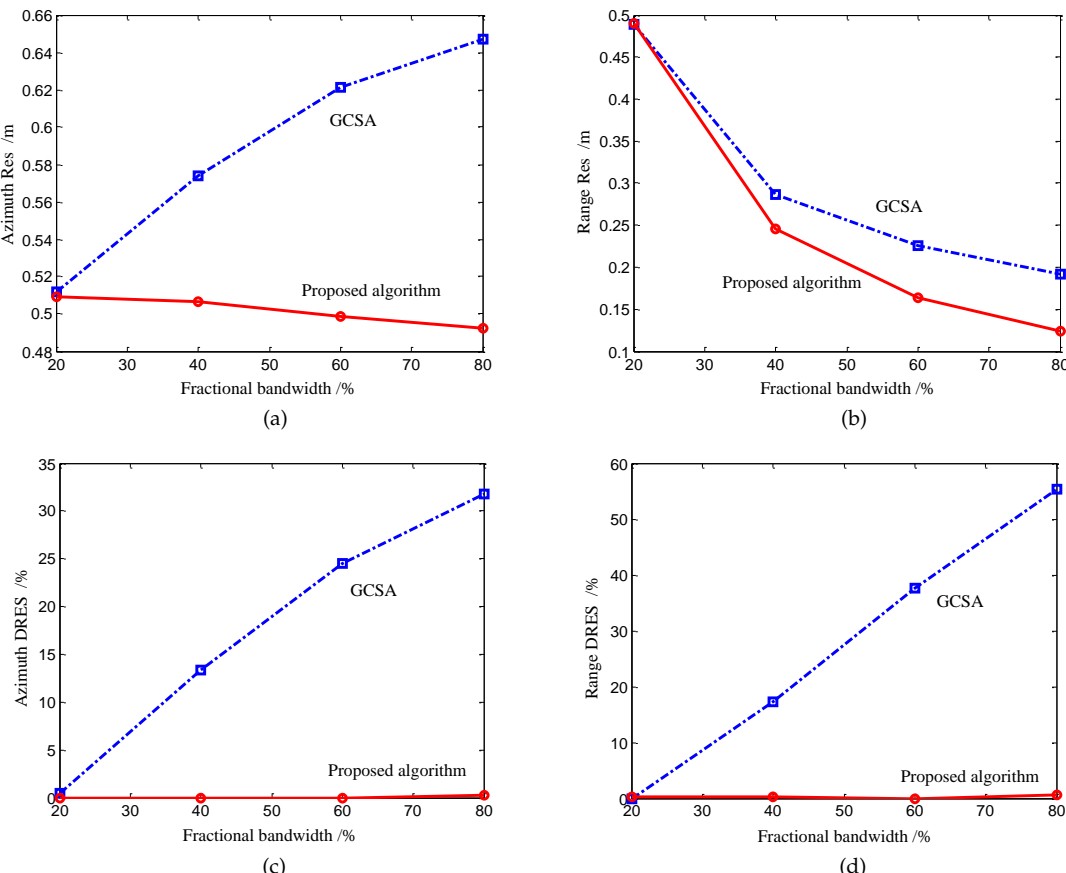

**Figure 8.** Resolutions (Res) and differential resolutions (DRES) in azimuth and range given by the generalized chirp scaling algorithm (GCSA) and proposed algorithm, where the resolutions obtained by $\omega - k$ algorithm are references. (**a**) Azimuth Res. (**b**) Range Res. (**c**) Azimuth DRES. (**d**) Range DRES.

Next, to illustrate the worst case shape of the proposed image of the point target, Figure 9 illustrates the contour plots, range profiles and azimuth profiles of the processed images at 80% fractional bandwidth. In both figures, the contour maps denote the 2D focusing quality and the profiles represent the focusing quality along the azimuth and range directions. The measured parameters are shown in Table 3. As can be seen, the results of GCSA in this case suffers form severe distortion and broadening. The signal in both range and azimuth are almost defocused. However, the proposed algorithm preserves the focusing performance. It is easy to recognize that the images obtained by the proposed algorithm are well focused, as are shown in Figure 9d–f. The nearly theoretical values of spatial resolution, PSLR and ISLR are obtained, which demonstrates the validity of the proposed algorithm. The good performance is given by the high-order range-independent phase filtering and the Lagrange inversion, which greatly reduces the range-dependent phase error. By comparing the range and azimuth profiles, it is evident that the Lagrange inversion makes the range-dependent coupling terms effectively compensated. The focusing depth in range dimension

is greatly improved. The improved algorithm is consistent with the conventional GCSA in terms of computational complexity but the focusing performance is significantly improved. The improved method provides an attractive solution for processing the low frequency large bandwidth and wide beamwidth SAR data.

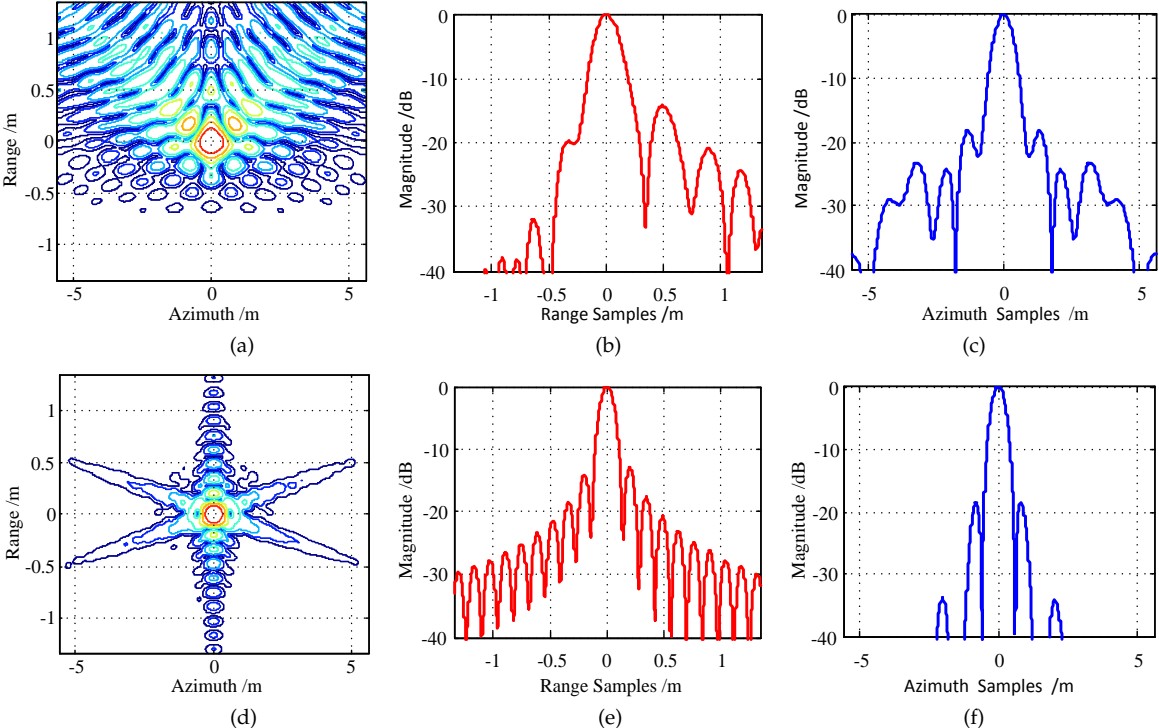

**Figure 9.** Focused results of point scatterers for L-band SAR data at 80% fractional bandwidth, using the generalized chirp scaling algorithm (GCSA) and proposed algorithm. (**a**–**c**) Conventional GCSA. (**d**–**f**) Proposed algorithm. The three subgraphs of each row correspond to the contour maps, range profiles and azimuth profiles, respectively. The dynamic range of contour is −35 dB∼0 dB.

**Table 3.** Measured parameters of imaging results for Figure 9.

| Method | Azimuth | | | Range | | |
|---|---|---|---|---|---|---|
| | Res /m | PSLR /dB | ISLR /dB | Res /m | PSLR /dB | ISLR /dB |
| Conventional GCSA in Reference [26] | 0.6471 | −18.3522 | −12.7106 | 0.1915 | −14.2602 | −9.0208 |
| Proposed algorithm | 0.4922 | −18.5128 | −16.9421 | 0.1239 | −12.9655 | −9.5501 |

Finally, to further test the analysis presented in this paper, a SAR real image (Longmen, Henan, China) is used as the input radar cross section to generate SAR echo. The center frequency of the transmitted signal is 400 MHz, the bandwidth is 250 MHz, the beamwidth is 25°, the velocity is 120 m/s, the PRF is 200 Hz, the pulse duration is 10 us and the center slant range is 10 km. The scene size is 3.0 km in range and 1.5 km in azimuth. The resolutions are 0.53 m in range and 0.76 m in azimuth. In the simulation, the input image is a real complex image. Each cell of the complex image is treated as a point scatterer (this actually contains the target signal and noise). No additional noise was added during the simulation. Figure 10 shows the imaging results processed by the conventional GCSA and the improved algorithm.

As is shown in Figure 10, at the center range region, the focusing quality of the two images is nearly the same. However, at the near and far range region, the texture character of the image obtained by improved GCSA is clearer than that obtained by conventional GCSA. It is obvious that better focusing results are obtained by the proposed method. The image has very high quality. The roads,

rivers and farmland can be clearly distinguished. The edge scene along range dimension is well focused. It is evident that the range focusing depth is greatly improved.

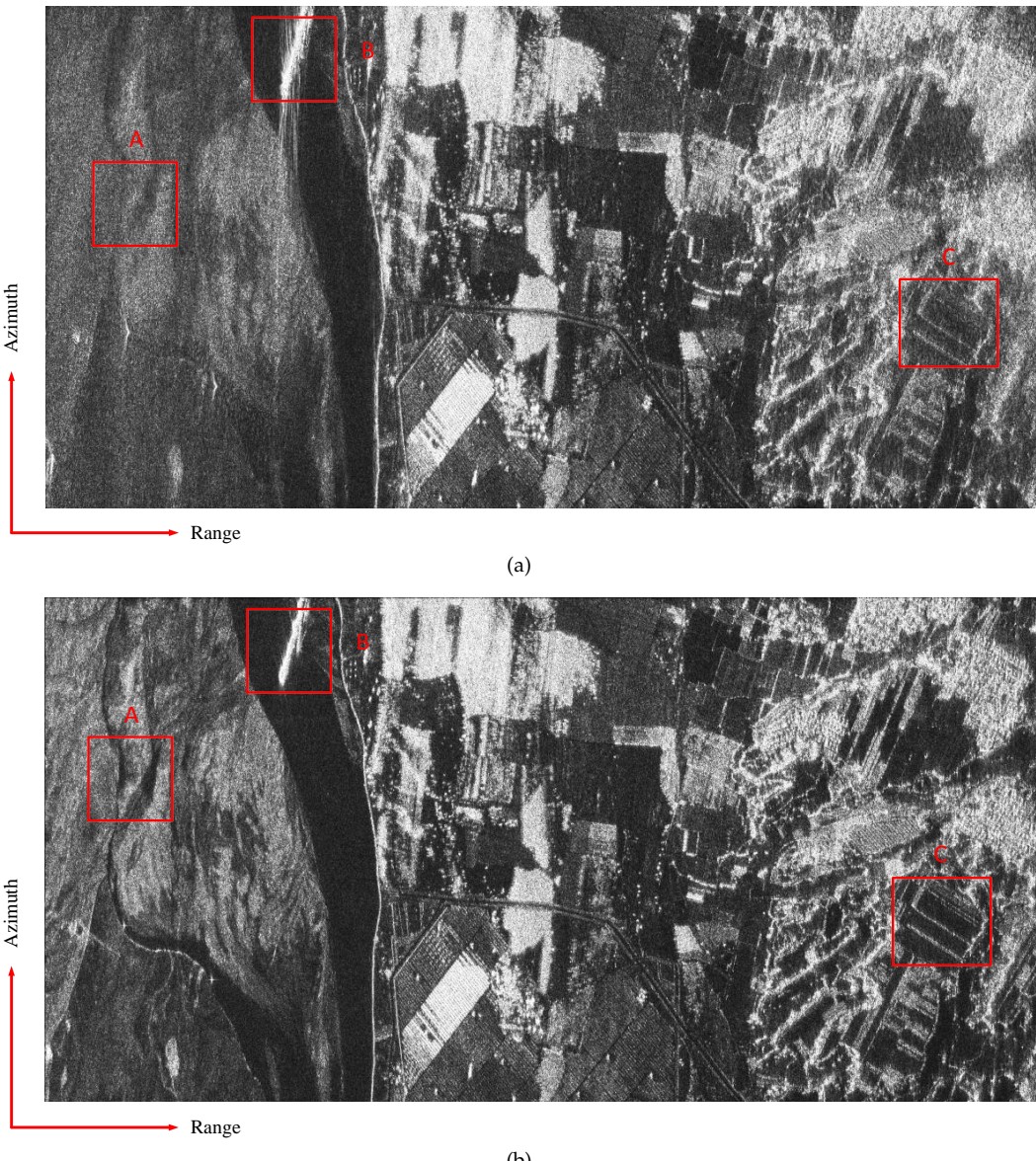

**Figure 10.** Comparison of imaging results of real data processed by different algorithms. (**a**) Conventional generalized chirp scaling algorithm (GCSA). (**b**) Proposed algorithm.

In order to have a distinct contrast, three subregions marked by red solid rectangle are extracted and analyzed in detail. The zooms regions are shown in Figure 11a–f. The entropy [33,34] of images is calculated to compare the focus quality and shown in Table 4. It is generally acknowledged that SAR images with better quality focus have smaller entropy. It is easy to find that images obtained by proposed method have smaller entropy. Therefore, the effectiveness of proposed algorithm is again validated.

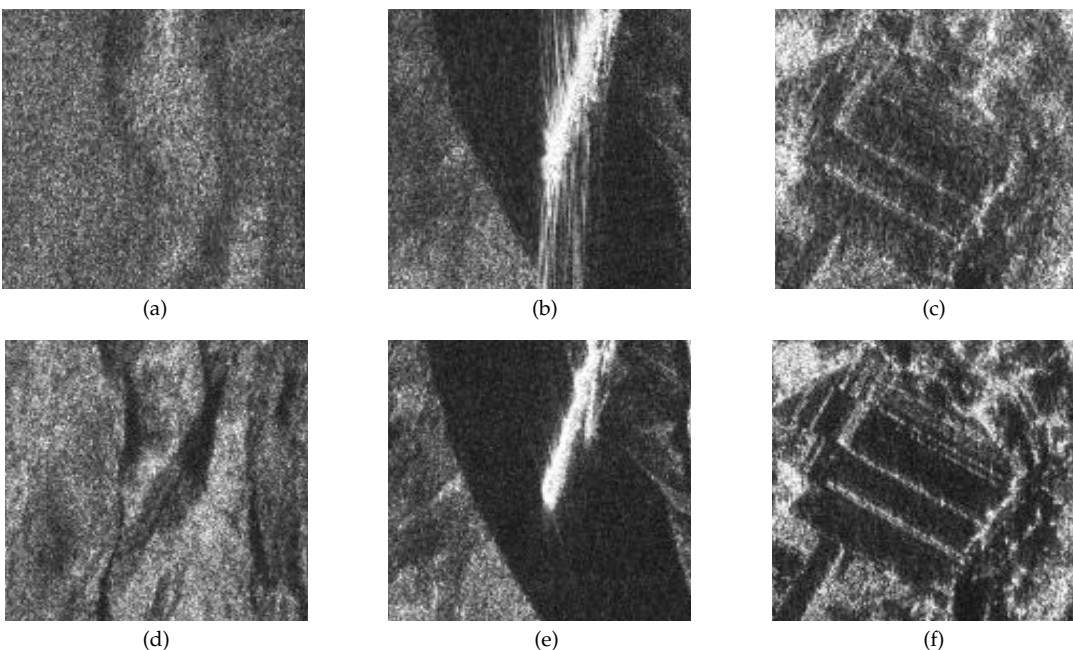

**Figure 11.** Zoom images. (**a**–**c**) Zoom images of subregions extracted from Figure 10a. (**d**–**f**) Zoom images of subregions extracted from Figure 10b.

**Table 4.** Image entropy of the zoom images.

| Method | Region A | Region B | Region C |
|---|---|---|---|
| Conventional GCSA in Reference [26] | 7.6037 | 7.1572 | 7.6275 |
| Proposed algorithm | 7.5957 | 6.9543 | 7.2874 |

## 5. Discussion

The performance of the proposed algorithm is mainly limited by the approximation error of range FM rate $K_m$ (Equation (22)). This approximation will introduce a quadratic phase error (QPE) in the spectrum and degrades the quality of the image. In the CSA, the range dependence of SRC is neglected and the FM rate is calculated at the reference range $K_{m,app} = K_f$. The NCSA uses a linear approximation of the FM rate and the GCSA uses a 2nd order approximation model.

The QPE can be expressed as

$$\Phi_{QPE}(f_\tau, f_\eta; R_0) = \pi f_\tau^2 \left( \frac{1}{K_m} - \frac{1}{K_{m,app}} \right) \tag{36}$$

From Equation (8), we can see that the Taylor expansion is feasible only under the following condition

$$\left| \frac{K_r c R_0 f_\eta^2}{2 V_r^2 f_0^3 D(f_\eta)^3} \right| \neq 1 \tag{37}$$

Let $G(K_r, R_0, V_r, f_0, f_\eta) = \frac{K_r c R_0 f_\eta^2}{2 V_r^2 f_0^3 D(f_\eta)^3}$, this function is positively related to $K_r$, $R_0$, $f_\eta$ and negatively related to $V_r$, $f_0$. And $K_r = B_r/T_r$, where $T_r$ is the pulse duration. The azimuth frequency varies within the following range $-\frac{PRF}{2} + f_{\eta c} \leq f_\eta \leq f_{\eta c} + \frac{PRF}{2}$, where PRF is the pulse repetition frequency. Thus, $G$ is an even function about $f_\eta$ and inequality (37) actually implied condition $G < 1$. This inequality is satisfied in most L- and P-band SAR systems. However, as the center frequency decreases, the maximum azimuth frequency increases and the slant range increases,

this inequality may not be satisfied. At this time, the Taylor approximation of range FM rate will introduce a large error and worsen the performance of algorithm.

It should be noted that $G < 1$ is required for all frequency domain approximation algorithms (such as the chirp scaling class and the range-Doppler class algorithms) [14]. This is satisfied for a linear FM signal with a large time bandwidth product (TBP). The TBP is defined as the product of the pulse duration $T_r$ and bandwidth $B_r$. For example, assuming the SAR system with a following parameters: center frequency $f_0 = 400$ MHz, bandwidth $B_r = 200$ MHz, beamwidth $\theta = 29°$, velocity $V_r = 100$ m/s, pulse duration $T_r = 2$ us (TBP = 400), target slant range $R_0 = 12$ km and center slant range $R_c = 10$ km. Figure 12a shows the variation of $G$ with azimuth frequency $f_\eta$. Figure 12c–e show the QPEs in zero-order, 1st-order and 2nd-order approximations, respectively. It can be seen that $G$ is not less than 1 at this time. Due to the existence of breakpoints, the QPEs of 1st-order and 2nd-order approximations are larger than zero-order approximation. The maximum QPE of 2nd-order model is about 5000°, which will seriously deteriorate the image quality.

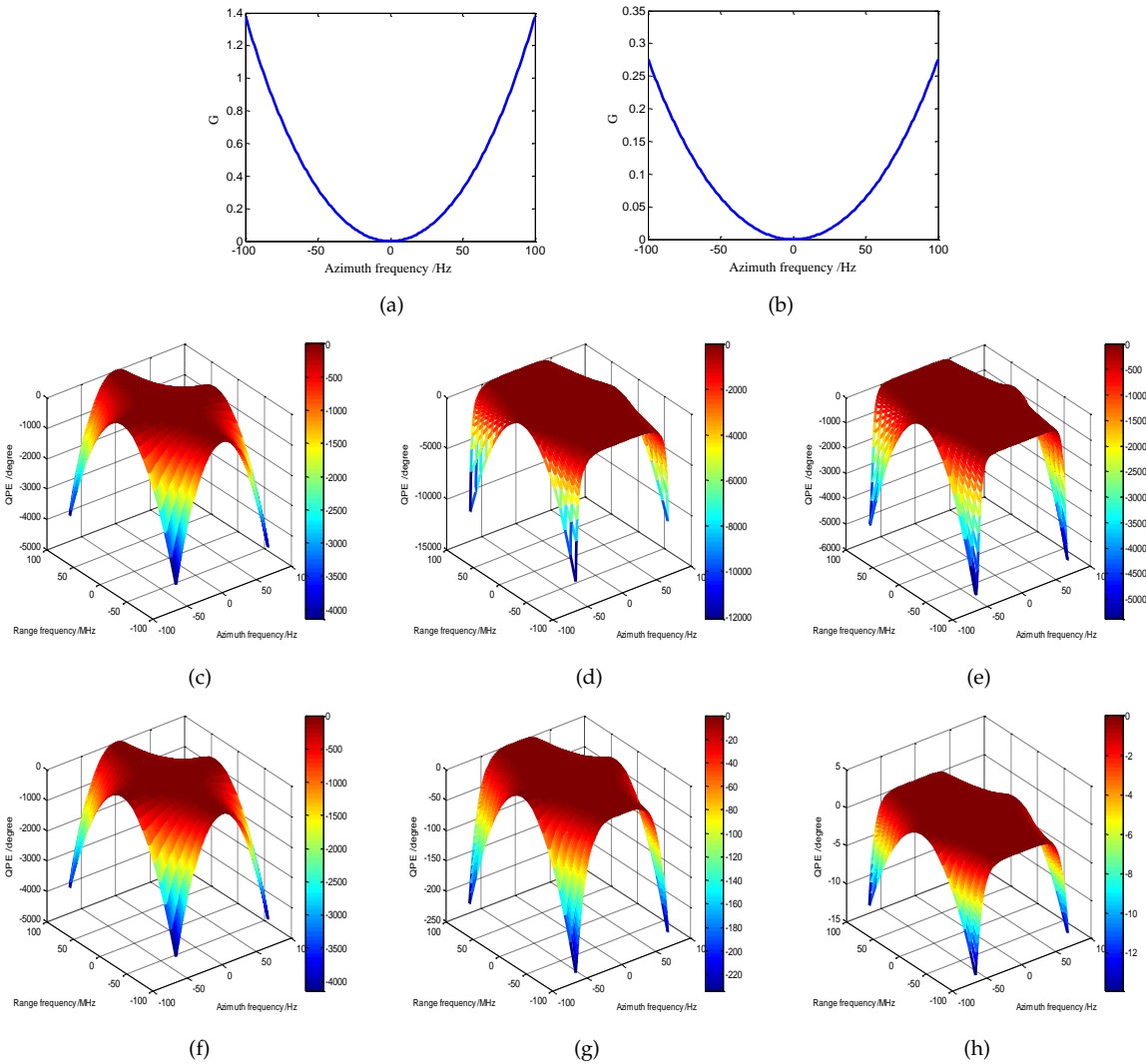

**Figure 12.** Parameters G and quadratic phase error (QPE). (**a**) Parameter G when $T_r = 2$ us (TBP = 400). (**b**) Parameter G when $T_r = 10$ us (TBP = 2000). (**c–e**) QPEs of zero-order, 1st-order and 2nd-order approximations corresponding to (**a**). (**f–h**) QPEs of zero-order, 1st-order and 2nd-order approximations corresponding to (**b**).

Change the pulse duration $T_r$ = 10 us (TBP = 2000), the variation of $G$ with azimuth frequency is show in Figure 12b. The QPEs of different approximations are also shown in Figure 12f–h. $G$ is far less than 1, meeting the inequality (37). At this time, the maximum QPE of 1st-order model is 230° and the maximum QPE of 2nd-order model is 14°, which has a limit effect on the imaging results.

By comparison, it can be concluded that the proposed algorithm has the best performance in frequency domain approximation algorithms when Equation (37) is satisfied. If Equation (37) is not satisfied, the performance of the approximation algorithm is degraded and the $\omega - k$ algorithm and time-domain algorithms are a better choice. At the same time, it should be pointed out that most of the existing airborne SAR systems have a large TBP [35,36], which shows that the proposed algorithm still has a broad application prospects. In particular, the proposed algorithm can be applied to high-resolution highly squint SAR imaging, where the phase coupling is also severe.

## 6. Conclusions

The high-resolution low frequency SAR has the characteristics of large bandwidth and long integration time. This trait will cause serious range-azimuth phase coupling, which limits the performance of conventional GCSA and results in image defocusing. The longer the integration time or larger bandwidth is, the more serious the deterioration will be. This paper proposes an improved GCSA based on Lagrangian inversion theorem for high-resolution low frequency SAR data processing.

Through the theoretical analysis, we find two main reasons about the defocusing. Firstly, the influence of the residual high-order phase is still significant when the fractional bandwidth is large and/or integration time is long. Secondly, the linear approximation of stationary phase point will make the high-order range-dependent phase coupling not effectively compensated. Aim to solve these two problems, this paper firstly proposes a new criterion for determining the order of Taylor expansion. The range-independent coupling phase terms above 3rd order are first compensated. Moreover, the Lagrange inversion theorem is introduced to obtain a more accurate stationary phase point. The performance and accuracy of the improved GCSA has been demonstrated using the simulated data in P- and L-band. The experimental results show that for P-band SAR systems with resolutions of 0.44 m, the proposed algorithm can focused full-swath targets with a resolution loss of less than 1%. For L-band SAR system with an azimuth resolution of 0.5 m, the edge point scatterer can be focused well even at 80% fractional bandwidth. The proposed algorithm has similar performance to the $\omega - k$ algorithm and is significantly better than the GCSA.The improved method provides the possibility to efficiently process full-swath high-resolution low frequency SAR data.

**Author Contributions:** All authors have made great contributions to the work. X.C. and T.Y. carried out the theoretical framework. X.C., T.Y., Z.H. and F.H. conceived and designed the experiments; X.C. performed the experiments and wrote the manuscript; X.C. and T.Y. analyzed the data; Z.D. gave insightful suggestions for the work and the manuscript.

**Acknowledgments:** This research was supported by the National Natural Science Found of China under Grant 61771478. The authors would like to thank all the anonymous reviewers for their valuable comments and helpful suggestions which lead to substantial improvements of this paper.

**Conflicts of Interest:** The authors declare no conflict of interest.

## Appendix A

Here, some expressions are given for some characteristics of the SAR signal. Using the Taylor expansion principle, the coefficients $\gamma_i$ in Equation (2) can be expressed as

$$
\begin{aligned}
\gamma_3 &= -\frac{D^2(f_\eta)-1}{2D^5(f_\eta)f_0^3} \\
\gamma_4 &= -\frac{(D^2(f_\eta)-1)(D^2(f_\eta)-5)}{8D^7(f_\eta)f_0^4} \\
\gamma_5 &= \frac{(D^2(f_\eta)-1)(3D^2(f_\eta)-7)}{8D^9(f_\eta)f_0^5} \\
\gamma_6 &= \frac{(D^2(f_\eta)-1)(D^4(f_\eta)-14D^2(f_\eta)+21)}{16D^{11}(f_\eta)f_0^6}
\end{aligned}
\tag{A1}
$$

**Appendix B**

In this Appendix, we will derive the calculation of $q_i (i \geq 2)$, $X_i (i \geq 3)$ and $C_i (i \geq 0)$.

Based on Equations (11), (17) and (20), we expand Equation (19) into a Taylor series with respect to $\tau_s$, the Equation (21) is obtained. The coefficient $C_0$ is a $M$th order polynomial of $\Delta\tau$ and it can be given by

$$
\begin{aligned}
C_0(\Delta\tau) = & \left[ \alpha^2 q_2 + (\alpha - 1)^2 K_m \right] \Delta\tau^2 + \left[ \alpha^3 q_3 + (\alpha - 1)^3 K_m^3 X_3 \right] \Delta\tau^3 \\
& + \frac{1}{8} \left[ 8\alpha^4 q_4 + 2(\alpha - 1)^4 K_m^4 \left( 9K_m X_3^2 + 4X_4 \right) - 16D(f_\eta) f_0 (\alpha - 1)^3 K_m^3 \gamma_3 \right] \Delta\tau^4 + \ldots
\end{aligned}
\tag{A2}
$$

In order to model the rang dependence of coefficients $C_i (i > 0)$, they are also approximated as a second order series in $\Delta\tau$,

$$
\begin{aligned}
C_1 = & 2\pi \left[ (\alpha - 1) K_m + \alpha q_2 \right] \Delta\tau + 3\pi \left[ \alpha^2 q_3 + (\alpha - 1)^2 K_m^3 X_3 \right] \Delta\tau^2 \\
C_2 = & \pi \left( K_m + q_2 \right) + 3\pi \left[ \alpha q_3 + (\alpha - 1) K_m^3 X_3 \right] \Delta\tau \\
& + \frac{3}{2}\pi \left[ 4\alpha^2 q_4 + (\alpha - 1) K_m^3 \left( 9 (\alpha - 1) K_m^2 X_3^2 + 4 (\alpha - 1) K_m X_4 - 4D(f_\eta) f_0 \gamma_3 \right) \right] \Delta\tau^2 \\
C_3 = & \pi \left( q_3 + K_m^3 X_3 \right) + \pi \left[ 4\alpha q_4 + (\alpha - 1) K_m^4 \left( 9K_m X_3^2 + 4X_4 \right) - 2D(f_\eta) f_0 K_m^3 \gamma_3 \right] \Delta\tau \\
& + \frac{1}{8}\pi \left[ 80\alpha^2 q_5 + 20(\alpha - 1)^2 K_m^5 \left( 27K_m^2 X_3^3 + 24K_m X_3 X_4 + 4X_5 \right) \right. \\
& \left. - 32D(f_\eta) f_0 (\alpha - 1) K_m^4 \left( 9K_m X_3 \gamma_3 + 2\gamma_4 \right) \right] \Delta\tau^2 \\
& \ldots
\end{aligned}
\tag{A3}
$$

For each $C_i$, the higher order terms of $\Delta\tau$ are very small and can be neglected. Combining Equation (22) and (A3), the coefficients $C_i (i > 0)$ can be rewritten as

$$
\begin{aligned}
C_1 & = C_{11}\Delta\tau + C_{12}\Delta\tau^2 \\
C_2 & = C_{20} + C_{21}\Delta\tau + C_{22}\Delta\tau^2 \\
C_3 & = C_{30} + C_{31}\Delta\tau + C_{32}\Delta\tau^2 \\
& \ldots \\
C_M & = C_{M0} + C_{M1}\Delta\tau + C_{M2}\Delta\tau^2
\end{aligned}
\tag{A4}
$$

with

$$
\begin{aligned}
C_{11} = & 2 \left[ \alpha q_2 + K_f (\alpha - 1) \right] \\
C_{12} = & 3\alpha^2 q_3 + K_f^2 (\alpha - 1) \left[ 2K_s + 3K_f (\alpha - 1) X_3 \right] \\
C_{20} = & K_f + q_2 \\
C_{21} = & 3\alpha q_3 + K_f^2 \left[ K_s + 3K_f (\alpha - 1) X_3 \right] \\
C_{22} = & \frac{1}{2} \left[ 12\alpha^2 q_4 + K_f^3 (18K_f K_s (\alpha - 1) X_3 + 27K_f^2(\alpha - 1)^2 X_3^2 + 2(K_s^2 + 6K_f(\alpha - 1)^2 X_4) \right. \\
& \left. - 12D(f_\eta) f_0 (\alpha - 1) \gamma_3) \right] \\
C_{30} = & q_3 + K_f^3 X_3 \\
C_{31} = & 4\alpha q_4 + 3K_f^4 K_s X_3 + K_f^4 (\alpha - 1) \left( 9K_f X_3^2 + 4X_4 \right) - 2D(f_\eta) f_0 K_f^3 \gamma_3 \\
C_{32} = & \frac{1}{8} \left[ 48K_f^5 K_s^2 X_3 + 20K_f^5(\alpha - 1)^2 \left( 27K_f^2 X_3^3 + 24K_f X_3 X_4 + 4X_5 \right) - 48K_f^4 K_s D(f_\eta) f_0 \gamma_3 \right. \\
& \left. + 80\alpha^2 q_5 + 8K_f^5 K_s (\alpha - 1) \left( 45K_f X_3^2 + 16X_4 \right) - 32D(f_\eta) f_0 K_f^4 (\alpha - 1) \left( 9K_f X_3 \gamma_3 + 2\gamma_4 \right) \right] \\
& \ldots
\end{aligned}
\tag{A5}
$$

According to (A4), the coefficients $C_i$ are range-dependent. The expression for these coefficients

contains linear and quadratic terms in $\Delta\tau$. To remove the range dependence of these terms, it is required to set the coefficients of $\Delta\tau$ to zero. Let $C_{11}$ in (A5) be zero, we can obtain

$$q_2 = K_f \frac{1-\alpha}{\alpha} \tag{A6}$$

Let $C_{12}$ and $C_{21}$ be zero, we can obtain

$$\begin{aligned} q_3 &= \frac{K_f^2 K_s (1-\alpha)}{3\alpha} \\ X_3 &= \frac{K_s (\alpha-2)}{3K_f (\alpha-1)} \end{aligned} \tag{A7}$$

Similarly, $C_{M-1,2}$ and $C_{M1}$ be zero, the expressions of $q_M$ and $X_M$ can be obtained. And the coefficients $C_i$ becomes

$$\begin{aligned} C_1 &= 0 \\ C_2 &= C_{20} \\ C_3 &= C_{30} \\ &\dots \\ C_M &= C_{M0} \end{aligned} \tag{A8}$$

Thus, Equations (25) and (26) are obtained.

**Appendix C**

This Appendix is used to explain how to use the Lagrange inversion theorem to solve the stationary phase point. According to Equations (12) and (13), $\tau$ can be expressed as a series form of $f_\tau$. A Lagrange inversion expression for a general power series is given here. For a function expressed in a series

$$\omega = h(z) = a_0 + a_1(z - z_0) + \dots + a_n(z - z_0)^n \tag{A9}$$

$$f(z) = \frac{z - z_0}{h(z) - a_0} = \frac{1}{a_1 + a_2(z - z_0) + \dots + a_n(z - z_0)^{n-1}} \tag{A10}$$

Thus, we can get

$$g_1 = f(z)|_{z=z_0} = \frac{1}{a_1} \tag{A11}$$

$$g_2 = \frac{1}{2} \frac{df^2(z)}{dz}\bigg|_{z=z_0} = -\frac{a_2}{a_1^3} \tag{A12}$$

$$g_3 = \frac{1}{6} \frac{d^2 f^3(z)}{dz^2}\bigg|_{z=z_0} = \frac{2a_2^3}{a_1^5} - \frac{a_3}{a_1^4} \tag{A13}$$

$$g_4 = \frac{1}{24} \frac{d^3 f^4(z)}{dz^3}\bigg|_{z=z_0} = -\frac{5a_2^3 - 5a_1 a_2 a_3 + a_1^2 a_4}{a_1^7} \tag{A14}$$

$$g_5 = \frac{1}{120} \frac{d^4 f^5(z)}{dz^4}\bigg|_{z=z_0} = \frac{14a_2^4 - 21a_1 a_2^2 a_3 + 3a_1^2 a_3^2 + 6a_1^2 a_2 a_4 - a_1^3 a_5}{a_1^9} \tag{A15}$$

$$g_6 = \frac{1}{720} \frac{d^5 f^6(z)}{dz^5}\bigg|_{z=z_0} = \frac{1}{a_1^{11}}\left(-42a_2^5 + 84a_1 a_2^3 a_3 - 28a_1^2 a_2^2 a_4 + 7a_1^2 a_2\left(-4a_3^2 + a_1 a_5\right) + a_1^3\left(7a_3 a_4 - a_1 a_6\right)\right) \tag{A16}$$

Equations (12) and (27) are special forms of Equation (A9) and the inversion expressions can be easily obtained according to Equation (14).

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
