# Peer review of "An Improved Generalized Chirp Scaling Algorithm Based on Lagrange Inversion Theorem for High-Resolution Low Frequency Synthetic Aperture Radar Imaging"

_remotesensing, doi:10.3390/rs11161874_

Round 1
Reviewer 1 Report
The paper entitled “An Improved Generalized Chirp Scaling Algorithm Based on Lagrange
Inversion Theorem for High-resolution Low Frequency SAR Imaging Journal: Remote Sensing” proposed an advanced chirp scaling algorithm with lagrange inversion and high-order reference phase compensation based on the simulation results. The results of simulation was interesting. However, the reviewer requires revisions following below points.
# major revision points.
What is the definition of High-resolution ? How did you determine the simulation parameter(Table 1)?
Through the entire of this paper, abstract expression was frequently used (e.g., “greatly”, “good”, “more accurate”, “not very large” and so on, ) without quantitative discussion and uncertainty quantification and sufficient information of simulation parameters setting criteria /observation information.
Though you have discussion part as section 5, large part of section 4. “Experiment Results and Analysis” is occupied by abstract expression and qualitative discussion with lack of quantative description of simulation results. Could you quantify uncertainty or build an index to explain the results quantitively_
# minor revision points.
The most part of this manuscript lacks an description of unit of variables and values (Particulalry, section 2 and figures). Please add the description of the unit.
L32.
Please describe brief definition of “phase coupling” reviewing/referring to related study .
L61
Beam wdith -> beam width
L67
The objective of this study is described here by describing “more accurate” .
But does not describe compared with what? Did you discuss the accuracy quantitively in this paper?
Fuigure 6, 7
What is the meaning of color of conter? Unit
Please spell out of each abbreviation in each figures legend.
Some abbreviations suddenly appears without spelling out and then spelled out later. This made the paper very hard to read.
e.g., L129 POSP (spelled out on L196.), HOPC L172 (spelled out L188)
The reviewer strongly requires carful check and revision of the structure of this paper.
Figure 9. what is the spec of this SAR? How about S/N ratio ? can you add more detail of the observation information like table 1?
What is the land cover or AOI A-C?
Author Response
Response to Reviewer 1:
The paper entitled “An Improved Generalized Chirp Scaling Algorithm Based on Lagrange Inversion Theorem for High-resolution Low Frequency SAR Imaging Journal: Remote Sensing” proposed an advanced chirp scaling algorithm with Lagrange inversion and high-order reference phase compensation based on the simulation results. The results of simulation were interesting.
We sincerely thank the reviewer for appreciating our work.
What is the definition of High-resolution?Thank you for your comments to improve our manuscript.
Answer:
We are very sorry for our negligence of the definition of “high-resolution”, but this is very important. In the revised manuscript, we gave this definition in the first paragraph in Section 1 (Line 30-34): “A high-resolution low frequency SAR system refers to a SAR system which operates with a low frequency (P- or L-band) signal with a large fractional bandwidth (>0.2, i.e. the ultra-wideband SAR) and a wide antenna beamwidth (corresponding to high azimuth resolution in the decimeter regime). The fractional bandwidth is defined by the ratio of the signal bandwidth to the center frequency.”
In addition, in order to better explain this definition, a description of the advantages of high-resolution [R1, R2] and low frequency [R3] is added.
[R1] Reigber, A.; Scheiber, R.; Jager, M. Very-High-Resolution Airborne Synthetic Aperture Radar Imaging: Signal Processing and Applications. Pro. IEEE. 2013, 101(3),759-783
[R2] Xie, H.; An, D.; Huang, X. Spatial resolution analysis of low frequency ultrawidebeam-ultrawideband synthetic aperture radar based on wavenumber domain support of echo data. J. Appl. Remote Sens. 2015,9(1), 095033.
[R3] Moore, R. K. Microwave Remote Sensing ; Addison-Wesley Pub. Co. Advanced Book Program, 1999.
Refs. [R1], [R2], [R3] have been added as Refs. [1], [4], [2] in Reference Section and marked in red on the revised manuscript.
How did you determine the simulation parameter (Table 1)?We thank the reviewer for the constructive suggestion.
Answer:
It should be noted that according to your Question 3, in order to better compare the performance of the algorithms, we have made some adjustments. The modified Table 1 is as follows:
Table 1. SAR system simulation parameters. (New)
|
Parameters |
P-band |
L-band |
|
Center frequency (MHz) |
600 |
1360 |
|
Bandwidth (MHz) |
300 |
272/544/816/1088 |
|
Fractional bandwidth (%) |
50 |
20/40/60/80 |
|
Beamwidth ( ) |
29 |
11 |
|
Azimuth resolution (m) |
0.44 |
0.5 |
|
PRF (Hz) |
240 |
240 |
|
Velocity of platform (m/s) |
100 |
100 |
|
Pulse duration (us) |
10 |
10 |
|
Center slant range (km) |
10 |
10 |
The Table in the original manuscript is as follow:
Table 2. SAR system simulation parameters. (old)
|
Parameters |
P-band 1 |
P-band 2 |
L-band |
|
Center frequency (MHz) |
600 |
600 |
1360 |
|
Bandwidth (MHz) |
200 |
300 |
600 |
|
Fractional bandwidth (%) |
33 |
50 |
48 |
|
Beamwidth ( ) |
19 |
29 |
28 |
|
Azimuth resolution (m) |
0.66 |
0.44 |
0.22 |
|
PRF (Hz) |
200 |
240 |
500 |
|
Velocity of platform (m/s) |
100 |
100 |
100 |
|
Pulse duration (us) |
10 |
10 |
10 |
|
Center slant range (km) |
10 |
10 |
10 |
Notes on the adjustment of parameters:
1) The P-band system 1 is deleted. Because there is already a comparison of the performance of different algorithm with fractional bandwidth, there is no need to provide a different set of bandwidths for performance. This modification is acceptable because the bandwidth and beamwidth of P-band system 1 are both smaller than that of P-band system 2, thus, the algorithm that can process system 2 well is sure to be able to process system 1.
2) In the revised manuscript, to investigate the effects of two improvements in the proposed algorithm, a P-band SAR with a center frequency of 600 MHz was simulated. The fractional bandwidth is set to 50%. Nine targets are arranged in the illuminated scene along the azimuth center at different distances from the reference range with an interval of 200 m. The azimuth resolution is set approximately equal to the range resolution.
3) In the revised manuscript, the bandwidth of L-band SAR becomes four different bandwidths in order to compare the variation of algorithm performance with fractional bandwidth. The azimuth resolution is set to 0.5 m.
In the new manuscript, we introduced the settings of parameter in detail. (Page 12: Line 276-287; Page 14: Line 320-325):
1) Parameters , and are set to the same value;
2) The PRF is chosen to be less than two times of the maximum Doppler frequency ;
3) Parameters and can be obtained by the following equation:
Through the entire of this paper, abstract expression was frequently used (e.g., “greatly”, “good”, “more accurate”, “not very large” and so on) without quantitative discussion and uncertainty quantification and sufficient information of simulation parameters setting criteria /observation information.
Though you have discussion part as section 5, large part of section 4. “Experiment Results and Analysis” is occupied by abstract expression and qualitative discussion with lack of quantative description of simulation results. Could you quantify uncertainty or build an index to explain the results quantitively?
Answer:
Considering the Reviewer’s suggestion, we have introduced the differential resolutions (DRES) [R4] to evaluate the accuracy of proposed algorithm quantitively. The DRS presents the loss in spatial resolutions. It should be noted that the three indexes (Res, PSLR, ISLR) in the original manuscript are also a quantitative description of the imaging results. The setting criteria of parameters is discussed in Question 2. More observation information is given (see in Question 10).
[R4]: Vu, V. T.; Sjogren, T. K.; Ultrawideband Chirp Scaling Algorithm. IEEE Geosci. Remote Sens. Lett. 2010, 7(2),281–285.
Refs. [R4] have been added as Refs. [34] in Reference Section and marked in red on the revised manuscript.
4. The most part of this manuscript lacks an description of unit of variables and values (Particularly, section 2 and figures). Please add the description of the unit.Answer:
We are very sorry that we cannot accurately understand the meaning of this question. It seems that we have introduced the unit of parameters in the paper (especially in Table 1 of Page 12). Could you please explain the question clearly?
5. (L32) Please describe brief definition of “phase coupling” reviewing/referring to related study.
Answer:
According to the reviewer’s suggestion, we have described the definition of “phase coupling” (Page 2, line 40-41): The phase coupling means the coupling between the range and azimuth frequencies in the phase of SAR transfer function.
And more details are described in the second paragraph of Section 2.1 (Page 3, line 110-121).
6. (L61) Beam wdith -> beam width
Answer:
We are very sorry for our incorrect spelling in the manuscript. In the revised manuscript, we have carefully checked the spelling issues to avoid the mistakes.
(L67) The objective of this study is described here by describing “more accurate”. But does not describe compared with what? Did you discuss the accuracy quantitively in this paper?
Answer:
The accuracy of the propose algorithm is compared to the existing GCSA. In the revision version, we redefined this information (Line 84-85): “The aim of this study is to overcome the limits of GCSA and propose a more accurate approach than the GCSA for processing wide-swath, high-resolution low frequency SAR data.”
Besides, the accuracy of the proposed algorithm is discussed quantitively in Section 4 in the new manuscript.
8. Figure 6, 7, what is the meaning of color of contour? Unit, Please spell out of each abbreviation in each figures legend.
Answer:
Figure 6 and 7 show the two-dimensional contour plots of the focused result with a dynamic range of [-35 dB, 0dB]. According to the reviewer’s suggestion, we have given this information in the figure legend. And the abbreviations in each figures legend are spelled out.
9. Some abbreviations suddenly appear without spelling out and then spelled out later. This made the paper very hard to read. e.g., L129 POSP (spelled out on L196.), HOPC L172 (spelled out L188). The reviewer strongly requires carful check and revision of the structure of this paper.
Answer:
Sorry for some abbreviations spelling problems in the manuscript. We have carefully checked the spelling issues to avoid the same mistakes in the revised manuscript.
10. Figure 9. what is the spec of this SAR? How about S/N ratio? can you add more detail of the observation information like table 1? What is the land cover or AOI A-C?
Answer:
Considering the Reviewer’s suggestion, we have given more information like Table 1 (Page16, Line 351-358). In the simulation, the input image is a real complex image. Each cell of the complex image is treated as a point scatterer (this actually contains the target signal and noise). No additional noise was added during the simulation. The land cover is 3.0 km in range and 1.5 km in azimuth.
Other changes:
Line 2, the statements of “phase coupling between the range and azimuth dimensions” were corrected as “range-azimuth phase coupling”. Line 4, the statements of “generalized chirp scaling (GCS) algorithm” were corrected as “generalized chirp scaling algorithm (GCSA)”. Line 7, the statements of “The degradation is mainly caused by two reasons. One of them is the residual high-order coupling phase, and the other is the non-negligible error introduced by the linear approximation when calculating the point of stationary phase using the principle of stationary phase (POSP).” were corrected as “The degradation is mainly caused by two reasons: the residual high-order coupling phase and the non-negligible error introduced by the linear approximation of stationary phase point”.
Special thanks to you for your good comments.
We tried our best to improve the manuscript and made some changes in the manuscript. These changes will not influence the content and framework of the paper. And here we did not list the changes but marked in red in revised paper. According to the suggestion, we have sent our paper to the native English speaker for the further polishment. We appreciate for Reviewers’ warm work earnestly, and hope that the correction will meet with approval.
Once again, thank you very much for your comments and suggestions.

Reviewer 2 Report
The authors propose an improvement to the generalized chirp scaling algorithm in the case of high resolution and low frequency SAR system. The proposed approach is sound and seems providing some improvements in the focused SAR images with respect to other approaches.
I didn’t find critical issues in the technical contents, but the manuscript could be improved in the description of the mathematical derivation of the algorithm. In particular both Section 3.2 and Appendix A suffer of clearness.
Also the discussion of the applicability limits deserve more details.
I suggest moderate revisions according to my comments in the following.
Appendix A is really a bit confusing. It reports derivation of some coefficients that are not really directly connected. For instance eq. (A1) is related to coefficients in eq. (2); eq. (A4) is related to coefficients in eq. (10) and so on. In particular the it is quite hard to follow the conceptual flow leading to eq (23) from (18). It should be: eq (20) + [ eq (18) + eq(16) ] + eq (4) + eq (21) = A (6) & A(7). This procedure should be explained more clearly in the text. For instance, at page 9 the dependency of eq (20) from parameter Km is hidden and the following considerations about Km and eq (21) seem not related to the previous text.
Section “Discussion”: authors point out the limit of the proposed approach, which is basically related to the approximation of Km. However, apparently there is not an algorithmic solution to this issue, which can be solved only by using suitable system parameters. This is a serious limitation for the application of the proposed algorithm on generic dataset. Authors should add further comments on this: in particular, how do other algorithms perform in the same conditions / approximation? Is the proposed solution still the best performing one?
Some notations could be improved: phi_RD in eq (18) is different from phi_RD in eq (15) thus they should be named differently. The same holds for phi_2D in eq (29), eq (26) and eq (10).
Page 17, Eq. (32): D(fn) is not present in eq (7). In general it is not present in the products involving range frequency rate (Kr) as in eq (1) and eq (3).
Figures 2, 6, 7, 8, 11 need colourbar.
Author Response
Response to Reviewer 2:
The authors propose an improvement to the generalized chirp scaling algorithm in the case of high resolution and low frequency SAR system. The proposed approach is sound and seems providing some improvements in the focused SAR images with respect to other approaches. I didn’t find critical issues in the technical contents.
We sincerely thank the reviewer for appreciating our work.
But the manuscript could be improved in the description of the mathematical derivation of the algorithm. In particular both Section 3.2 and Appendix A suffer of clearness. Also the discussion of the applicability limits deserves more details.
Thank you for your comments to improve our manuscript.
Appendix A is really a bit confusing. It reports derivation of some coefficients that are not really directly connected. For instance eq. (A1) is related to coefficients in eq. (2); eq. (A4) is related to coefficients in eq. (10) and so on. In particular the it is quite hard to follow the conceptual flow leading to eq (23) from (18). It should be: eq (20) + [ eq (18) + eq(16) ] + eq (4) + eq (21) = A (6) & A(7). This procedure should be explained more clearly in the text. For instance, at page 9 the dependency of eq (20) from parameter Km is hidden and the following considerations about Km and eq (21) seem not related to the previous text.
We thank the reviewer for the constructive suggestion.
Answer:
According to the Reviewer’s suggestion, we have rewritten this part. Appendix A is split into two parts. In Appendix B, we present the detailed process for solving the parameters , and . Equations (A2)-(A4) are placed in Section 3.2 (Equation (11), (23) and (24) in the new manuscript), making the context more fluid. The implied relationship between Equation (21) and parameter is also mentioned (Page 10, line 250-252).
The above description has been added in Section 3.2 and Appendix B and marked in red in the revised manuscript.
Section “Discussion”: authors point out the limit of the proposed approach, which is basically related to the approximation of Km. However, apparently there is not an algorithmic solution to this issue, which can be solved only by using suitable system parameters. This is a serious limitation for the application of the proposed algorithm on generic dataset. Authors should add further comments on this: in particular, how do other algorithms perform in the same conditions / approximation? Is the proposed solution still the best performing one?
Answer:
We have made correction according to the Reviewer’s comments. In the revised manuscript, we pointed out that (Page 19, line 390-394): Parameter G<1 is required for all frequency domain approximation algorithms (such as the chirp scaling class and the range-Doppler class algorithms). This is satisfied for a linear FM signal with a large time bandwidth product (TBP).
(Page 20, line 405-411) By comparison, it can be concluded that the proposed algorithm has the best performance in frequency domain approximation algorithms when Equation (37) is satisfied. If Equation (37) is not satisfied, the performance of the approximation algorithm is degraded, and the algorithm and time-domain algorithms are a better choice. At the same time, it should be pointed out that most of the existing airborne SAR systems have a large TBP [R1, R2], which shows that the proposed algorithm still has a broad application prospects. In particular, the proposed algorithm can be applied to high-resolution highly squint SAR imaging, where the phase coupling is also severe.
[R1] Damini, A.; McDonald, M.; Haslam, G.E. X-band wideband experimental airborne radar for SAR, GMTI and maritime surveillance. IEE Proc.– Radar, Sonar and Navig. 2003, 150(4), 305–312.
[R2] Ender, J.H. G.; Brenner, A.R. PAMIR–a wideband phased array SAR/MTI system. IEE Proc.– Radar, Sonar and Navig. 2003, 150(3), 165–172.
Refs. [R1],[R2] have been added as Refs. [35],[36] in Reference Section and marked in red on the revised manuscript.
Some notations could be improved: phi_RD in eq (18) is different from phi_RD in eq (15) thus they should be named differently. The same holds for phi_2D in eq (29), eq (26) and eq (10).Answer:
Considering the Reviewer’s suggestion, we renumbered these Equations in order. And all the changes have been marked in red in the revised manuscript.
Page 17, Eq. (32): D(fn) is not present in eq (7). In general it is not present in the products involving range frequency rate (Kr) as in eq (1) and eq (3).
Answer:
We are very sorry for our incorrect writing. The migration factor is not present in Equation (32). We have made correction according to the Reviewer’s comments (Page 18).
Figures 2, 6, 7, 8, 11 need colour bar.
Answer:
Considering the Reviewer’s suggestion, we have added the colour bars to Figure 2 and 11 (new number is 12 in the revised manuscript). However, Figure 6-8 is the contour plots of the focused results. In most of the literature [R3, R4, R5], colour bars are not added to the contour map. In the new manuscript we pointed out that its dynamic range of the contour map is -35 dB to 0 dB. Is this enough?
Refs. [R3],[R4], [R5] are the Refs.[19], [22], [23] in the new manuscript.
Special thanks to you for your good comments.
We tried our best to improve the manuscript and made some changes in the manuscript. These changes will not influence the content and framework of the paper. And here we did not list the changes but marked in red in revised paper. We appreciate for Reviewers’ warm work earnestly, and hope that the correction will meet with approval.

Round 2
Reviewer 2 Report
Authors revised the manuscript by following my suggestions.
Now it is ready for publication.